# Exploration-Driven Optimization for Test-Time Large Language Model Reasoning

**Changhao Li** *cli911@gatech.edu*
*Georgia Institute of Technology*

**Yuchen Zhuang** *yczhuang@gatech.edu*
*Georgia Institute of Technology*

**Chenxiao Gao** *cgao@gatech.edu*
*Georgia Institute of Technology*

**Haotian Sun** *haotian.sun@gatech.edu*
*Georgia Institute of Technology*

**Rushi Qiang** *rqiang6@gatech.edu*
*Georgia Institute of Technology*

**Chao Zhang** *chaozhang@gatech.edu*
*Georgia Institute of Technology*

**Bo Dai** *bodai@cc.gatech.edu*
*Georgia Institute of Technology*

**Reviewed on OpenReview:** *https://openreview.net/forum?id=NiINDlzvNj*

## Abstract

Post-training techniques combined with inference-time scaling significantly enhance the reasoning and alignment capabilities of large language models (LLMs). However, a fundamental tension arises: inference-time methods benefit from diverse sampling from a relatively flattened probability distribution, whereas reinforcement learning (RL)-based post-training inherently sharpens these distributions. To address this, we propose Exploration-Driven Optimization (`EDO`), which extends reward-biasing style exploration objectives to iterative post-training and integrates them into standard RL objectives, encouraging greater diversity in sampled solutions while facilitating more effective inference-time computation. We incorporate `EDO` into iterative Direct Preference Optimization (iDPO) and Group Relative Policy Optimization (GRPO), resulting in two variants: `ED-iDPO` and `ED-GRPO`. Extensive experiments demonstrate that both `ED-iDPO` and `ED-GRPO` exhibit greater solution diversity and improved reasoning abilities, particularly when combined with test-time computation techniques like self-consistency. Across three in-distribution reasoning benchmarks, `EDO` achieves a 1.0-1.3% improvement over the strongest baselines, and delivers an additional 1.5% average gain on five out-of-distribution tasks. Beyond accuracy, `EDO` preserves model entropy and stabilizes RL training dynamics, highlighting its effectiveness in preventing over-optimization collapse. Taken together, these results establish `EDO` as a practical framework for balancing exploration and exploitation in LLM reasoning, especially in settings that rely on test-time scaling.

# 1 Introduction

Recent advances have shown the increasing effectiveness of post-training techniques for enhancing the reasoning capabilities of large language models (LLMs). State-of-the-art models like DeepSeek-R1 (Guo et al., 2025) and OpenAI's o1 (Jaech et al., 2024) have demonstrated substantial improvements across diverse reasoning tasks, including mathematical problem-solving, programming, and scientific reasoning. These gains underscore the important role of reinforcement learning (RL) in extending long chain-of-thought (Wei et al., 2022) reasoning capabilities. Additionally, recent studies (Liu et al., 2025; Zuo et al., 2025) suggest that the reasoning performance of LLMs can be further enhanced through computationally efficient inference-time scaling methods (Zhang et al., 2025a), which typically involve using reward models (Lightman et al., 2023; Zhang et al., b) to refine generated can-

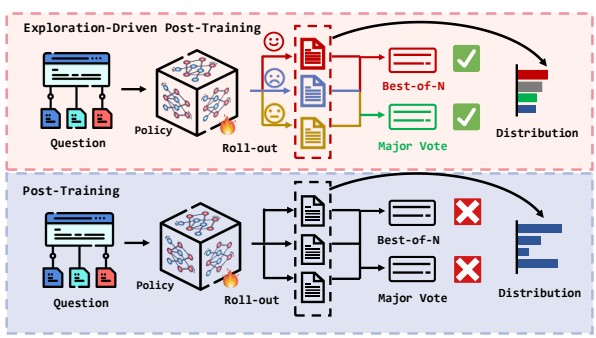

Figure 1: `EDO` exhibits a more flattened probability distribution, encouraging greater solution diversity and enhancing reasoning performance when integrated with inference-time computation techniques. 🔥 indicates the trainable parameters.

didates and subsequently aggregating multiple outputs via techniques such as Majority Voting (Wang et al., 2022), Best-of-N sampling (Stiennon et al., 2020), or Monte Carlo tree search (Hao et al., 2023). Despite these promising results, most existing approaches treat inference-time scaling as an *independent, post-hoc* procedure applied after standard post-training. However, separating the training process from inference-time scaling introduces a fundamental tension (Chow et al., 2024): effective inference-time scaling benefits from diverse sampling from *relatively flattened distributions*, whereas conventional RL-based post-training methods inherently sharpen output distributions by fine-tuning the model toward generating high-confidence, high-reward solutions (Kirk et al., 2023; Santurkar et al., 2023) (see Figure 1). Encouraging greater exploration during post-training could alleviate this tension, enabling the model to leverage diverse inference pathways and enhancing its ability to draw complex inferences and accurate solutions (Chen et al., 2024).

To this end, we propose Exploration-Driven Optimization (`EDO`), designed specifically to promote diversity during post-training. Unlike conventional RL objectives that directly optimize for high-reward responses, `EDO` builds on reward-biasing style exploration objectives from prior work (Cen et al., 2024; Zhang et al., 2024; Xie et al., 2024; Liu et al., 2023; 2024) and extends them to iterative post-training settings that interact naturally with test-time scaling. In this way, `EDO` simultaneously achieves strong alignment with task objectives while also encouraging the exploration of diverse output generations, thus facilitating the test-time inference naturally. To accommodate on-policy training, we extend `EDO` to leverage iterative self-improvement by sampling from the current checkpoint and using the earlier checkpoints as reference policies. We demonstrate that the design of `EDO` can be seamlessly integrated into existing post-training pipelines, including iterative Direct Preference Optimization (iDPO) and Group Relative Policy Optimization (GRPO), yielding two powerful variants: `ED-iDPO` and `ED-GRPO`. Extensive empirical evaluations across three reasoning benchmarks demonstrate that `EDO` consistently enhances both *in-distribution* and *out-of-distribution* performance, particularly when combined with inference-time computation strategies such as self-consistency (Wang et al., 2022). Notably, `EDO` achieves higher diversity metrics than other baselines under a fixed prompt set. Moreover, it is compatible with various policy optimization algorithms and test-time inference techniques, while consistently maintaining strong overall performance. In conclusion, we summarize our main contributions as follows:

- **i),** We introduce `EDO`, a principled optimization framework, to promote a flattened output distribution for exploration-exploitation balance, thus facilitating more effective inference-time scaling;
- **ii),** `EDO` integrates seamlessly with popular post-training paradigms, producing enhanced variants (`ED-iDPO` and `ED-GRPO`) that significantly improve solution diversity and reasoning performance in conjunction with inference-time strategies; and
- **iii),** Our empirical evaluation demonstrates that `EDO` yields consistent improvements across various backbones and datasets, highlighting `EDO`'s effectiveness and robustness for LLM post-training.

## 2 Preliminaries

**Reinforcement Learning for Large Language Models.** In recent years, RL has emerged as a powerful paradigm to align LLMs with human values or to elicit advanced capabilities from pretrained models. Standard RL assumes access to a reward function $r(x, y) : \mathcal{X} \times \mathcal{Y} \to \mathbb{R}$, which provides scalar feedback on a generated response $y \in \mathcal{Y}$ given a prompt $x \in \mathcal{X}$. The most popular formulation of RL optimizes the policy $\pi_\theta : \mathcal{X} \to \Delta(\mathcal{Y})$ with the following regularized objective:

$$J^*(r) = \max_{\pi_\theta} J(r, \pi_\theta) = \max_{\pi_\theta} \mathbb{E}_{x \sim \mathcal{D}, y \sim \pi_\theta(\cdot|x)} \left[ r(x, y) - \beta \, \mathbb{D}_{\mathrm{KL}} \left( \pi_\theta(\cdot \mid x) \, \| \, \pi_{\mathrm{ref}}(\cdot \mid x) \right) \right], \tag{1}$$

where $\beta$ is a regularization coefficient that controls the deviation from the reference policy, $\mathbb{D}_{\mathrm{KL}}$ denotes the Kullback-Leibler (KL) divergence, and $\pi_{\mathrm{ref}}$ is typically specified as the policy after supervised-finetuning (Rafailov et al., 2023) to prevent optimization collapse.

This framework naturally applies to scenarios where a ground-truth reward function is available, for example, in mathematics or programming tasks where the correctness of the solution can be automatically verified. However, in open-ended applications where such a reward signal is absent, a reward model must be learned from alternative forms of feedback. A common approach is to assume access to a preference dataset $\mathcal{D}_p$ consisting of triplets $(x, y_w, y_l)$, where $x$ is the prompt, $y_w$ and $y_l$ are two model responses, and $y_w$ is preferred over $y_l$ according to human annotators or automated heuristics. To relate these observed preferences to rewards, the Bradley-Terry model (Bradley & Terry, 1952) is typically employed. It specifies the probability of preferring $y_w$ over $y_l$ as

$$p(y_w \succ y_l \mid x) := \sigma(r(x, y_w) - r(x, y_l)) = \frac{\exp(r(x, y_w))}{\exp(r(x, y_w)) + \exp(r(x, y_l))}. \tag{2}$$

When a static dataset of preference-labeled data pairs $\mathcal{D}_p = \{x_i, y_{i,w}, y_{i,l}\}_{i=1}^N$ is available, a parameterized reward model $r_\phi$ can be learned via minimizing the negative log-likelihood of the preferences:

$$\mathcal{L}(r_\phi, \mathcal{D}_p) = -\mathbb{E}_{(x, y_w, y_l) \sim \mathcal{D}_p}[\log \sigma(r_\phi(x, y_w) - r_\phi(x, y_l))]. \tag{3}$$

The learned reward model $r_\phi$ can subsequently be used to provide feedback for LLM-generated responses within the standard reinforcement-learning framework, as described in Eq. 1.

**(Iterative) Direct Preference Optimization.** The regularized RL objective in Eq. 1 admits a closed-form optimal solution, $\pi^*(a \mid s) \propto \pi_{\mathrm{ref}}(a \mid s) \exp(r(s, a))$. DPO (Rafailov et al., 2023) exploits this connection by reparameterizing the reward model with $r(s, a) = \log \frac{\pi_\theta(a|s)}{\pi_{\mathrm{ref}}(a|s)} + C$ and substituting this into Eq. 3, thus directly optimizing the policy via maximizing the likelihood of preferences:

$$\mathcal{L}_{\mathrm{DPO}} = -\mathbb{E}_{(x, y_w, y_l) \sim \mathcal{D}} \left[ \log \sigma \left( \beta \log \frac{\pi_\theta(y_w|x)}{\pi_{\mathrm{ref}}(y_w|x)} - \beta \log \frac{\pi_\theta(y_l|x)}{\pi_{\mathrm{ref}}(y_l|x)} \right) \right]. \tag{4}$$

Building upon this, recent studies (Wu et al., 2024; Rosset et al., 2024) have adopted iterative variants of the DPO framework (iDPO) to gradually enhance model performance through multiple iterations. The policy is first initialized as $\pi_\theta^0 = \pi_{\mathrm{ref}}$ and then optimized using DPO. The updated policy is subsequently used to generate new responses, for which additional human preference annotations are collected. This sampling-and-optimization cycle is repeated iteratively, thus enabling continual refinement of both the policy.

**Group Relative Policy Optimization** Alternatively, Proximal Policy Optimization (PPO) (Schulman et al., 2017) optimizes Eq. 1 by subsuming the KL penalty into the reward $r(x, y) = r_\phi(x, y) - \beta(\log \pi_\theta(y|x) - \log \pi_{\mathrm{ref}}(y|x))$ and maximizing the following token-level objective:

$$J^{\mathrm{PPO}} = \mathbb{E}_{x \sim \mathcal{D}, y \sim \pi_{\mathrm{old}}(\cdot|x)} \left[ \frac{1}{|y|} \sum_{t=1}^{|y|} \min \left( \frac{\pi_\theta(y_t|x, y_{<t})}{\pi_{\mathrm{old}}(y_t|x, y_{<t})} A_t, \ \mathrm{clip} \left( \frac{\pi_\theta(y_t|x, y_{<t})}{\pi_{\mathrm{old}}(y_t|x, y_{<t})}, 1 - \epsilon, 1 + \epsilon \right) A_t \right) \right], \tag{5}$$

where $A_t$ denotes the advantage values estimated using *Generalized Advantage Estimation* (GAE) (Schulman et al., 2015) based on the reward $r$ and a learned value function $V_\psi$. The policy $\pi_{\mathrm{old}}$ is the behavior model that is used to sample the responses in each iteration. Nevertheless, computing the value function in PPO

typically requires training a separate model comparable in size to the policy, which not only introduces substantial computational overhead but also introduces the risk of estimation error. To address this, Group Relative Policy Optimization (GRPO) instead estimates the advantage using *group-wise normalized rewards* $\hat{A}_{i,t} = r_i - \mathrm{mean}(\{r_{i'}\}_{i'=1}^G)$. Additionally, GRPO practices the KL regularization during policy optimization, rather than reward calculation. Collectively, these modifications lead to the following objective:

$$J^{\mathrm{GRPO}} = \mathbb{E}_{x \sim \mathcal{D}, \{y_i\}_{i=1}^G \sim \pi_{\mathrm{old}}(\cdot|x)} \bigg[ \tag{6}$$

$$\frac{1}{G}\sum_{i=1}^{G}\frac{1}{|y_i|}\sum_{t=1}^{|y_i|}\bigg(\min\bigg(\frac{\pi_\theta(y_{i,t}|x,y_{i,<t})}{\pi_{\mathrm{old}}(y_{i,t}|x,y_{i,<t})}\hat{A}_{i,t},\ \mathrm{clip}\bigg(\frac{\pi_\theta(y_{i,t}|x,y_{i,<t})}{\pi_{\mathrm{old}}(y_{i,t}|x,y_{i,<t})},1-\epsilon,\,1+\epsilon\bigg)\hat{A}_{i,t}\bigg)\ -\ \beta\,\mathbb{D}_{\mathrm{KL}}\big(\pi_\theta\,\|\,\pi_{\mathrm{ref}}\big)\bigg)\bigg].$$

## 3  Related Work

**Reinforcement Learning for LLM Reasoning**  Reinforcement learning (RL), especially Reinforcement Learning with Verifiable Rewards (RLVR), has become a core paradigm for improving LLM reasoning (Patil, 2025; Li et al., 2024; Xu et al., 2025). RLVR denotes RL methods where rewards are produced by deterministic, automatically checkable signals, such as math verifiers, program tests, or symbolic solvers, rather than subjective human preferences. This provides low-noise, scalable supervision that is especially effective for multi-step reasoning tasks. More broadly, RL for LLMs has traditionally been framed as Reinforcement Learning from Human Feedback (RLHF), where policies are optimized using human preference data or learned reward models (Ouyang et al., 2022; Jiang et al., 2024). Actor-critic methods like PPO (Schulman et al., 2017) are widely adopted due to their training stability, though they require extensive hyperparameter tuning to avoid overoptimization or reward hacking (Hochlehnert et al., 2025). Several recent variants aim to mitigate these challenges. VinePPO (Kazemnejad et al., 2024) replaces learned value functions with Monte Carlo returns obtained from rollouts, while VAPO (Yuan et al., 2025) and DAPO (Yu et al., 2025) are designed to correct biases inherent in value-based learning. GRPO (Shao et al., 2024) eliminates the critic entirely by leveraging direct comparisons among sampled responses, reducing computational complexity. DPO (Rafailov et al., 2023) further simplifies the RLHF pipeline by framing preference optimization as a purely supervised learning objective that bypasses reward modeling and policy gradient updates. Recent work has also addressed the diversity limitations of DPO-style alignment losses. In particular, SPO (Sharifnassab et al., 2024) adds a regularization term over the model's full output distribution to avoid overly sharp aligned policies. This is related in spirit to our work, since both aim to mitigate distributional collapse. The main difference is that SPO focuses on regularizing preference optimization itself, while EDO introduces an iterative exploration regularizer relative to the previous policy and studies its benefit for test-time scaling. Although these approaches substantially improve the efficiency and stability of policy optimization, maintaining sufficient output diversity for downstream reasoning and test-time scaling remains relatively underexplored. Such distributional collapse reduces sampling diversity and can impede reasoning, exploration, and test-time scaling (Cui et al., 2025; GX-Chen et al., 2025). Our work, EDO, directly addresses this limitation by encouraging broader exploration and maintaining more diverse solution distributions during optimization.

**Self-Improvement for LLM Reasoning**  Fine-tuning techniques have been widely used to enhance the mathematical reasoning abilities of LLMs (Yu et al., 2023; Luo et al., 2023; Yue et al., 2023). However, their effectiveness largely depends on high-quality, annotated question-response pairs featuring detailed chain-of-thought reasoning, resources that are becoming increasingly scarce and costly. To mitigate reliance on extensive human annotations, self-improvement approaches have emerged as promising alternatives. Broadly, these self-improvement methods fall into two main categories: (1) *Training on self-generated data*, where recent efforts (Gulcehre et al., 2023; Dou et al., 2024; Yuan et al., 2023) identify high-quality responses from model-generated outputs to iteratively fine-tune the LLM. Others (Setlur et al., 2024; Li et al., 2024) leverage both positive and negative responses to update the policy model via direct preference optimization (Rafailov et al., 2023); and (2) *Augmenting models with test-time computation*, involving strategies that enable LLMs to self-reflect or verify their outputs (Weng et al., 2022; Guo et al., 2025), typically through the generation of more extensive reasoning sequences. Alternative methods adjust model predictions based on feedback signals from either external environments or dedicated critic modules (Shinn et al., 2023; Zhou et al., 2023). EDO integrates self-improvement mechanisms at both training and inference phases, effectively leveraging the strengths of both strategies to enhance overall reasoning performance.

# 4 Method

In this section, we introduce `EDO`, an exploration-driven optimization framework designed to reduce the sharpness of the generation probability distribution via RL post-training. To achieve this, we first reinterpret test-time compute as an implicit Q-function that adaptively guides the generation process at inference time (Section 4.1). Building on this foundation, we propose a novel *Exploration-Driven Optimization* mechanism (Section 4.2) that explicitly softens the token probability distribution to promote broader exploration. This mechanism can be seamlessly integrated with two widely adopted online RL algorithms: Iterative Direct Preference Optimization (iDPO) and Group Relative Policy Optimization (GRPO), resulting in consistently improved generation quality and diversity. A detailed overview of the full `EDO` framework is provided in Section 4.3, and its empirical advantages are demonstrated in Section 5.2.

## 4.1 Test-time Computation as Q-function

At the inference time, given an input prompt $x$, we first generate a candidate pool $\mathcal{Y}_x = \{y_1, y_2, \cdots, y_N\}$ by sampling $N$ responses from the policy model $\pi_\theta$. The final response $\hat{y}$ is then selected from $\mathcal{Y}_x$ using a value estimation method. For instance, the majority voting scheme, expressed as $\hat{y} = \arg\max_{c \in C} \sum_{i=1}^{N} \mathbb{I}(y_i \Rightarrow c)$ , selects the candidate appearing most frequently, where $\mathbb{I}(y_i \Rightarrow c)$ denotes the indicator function specifying whether candidate response $y_i$ yields the final answer $c$ from a set of possible answers $C$. Alternatively, the best-of-$N$ sampling approach, defined as $\hat{y} = \arg\max_{y \in \mathcal{Y}_x} R(x, y)$, employs a reward model $R(x, y)$ to assign scores to each candidate $y \in \mathcal{Y}_x$ and selects the one with the highest reward. Under this formulation, we can frame candidate generation as the policy model's exploration of the solution space, while the selection stage functions as an *implicit* Q-function guiding towards the most promising candidate. In addition, we also provide an *explicit* Q-function using a tree-based search algorithm guided by a process reward model, as detailed in Appendix I.

Recent studies have demonstrated the effectiveness of test-time inference as an implicit form of Q-guidance in the context of Best-of-N (BoN) sampling (Gui et al., 2024; Yang et al., 2024b). Their findings highlight that policy models must engage in sufficient exploration during generation, as limited initial sampling restricts the discovery of optimal responses. Building on this insight, we propose **Exploration-Driven Optimization**, a framework that explicitly encourages broader exploration within the policy model (Section 4.2), accompanied by a detailed description of the complete training pipeline underlying this exploration-driven paradigm (Section 4.3).

## 4.2 Exploration-Driven Optimization

In post-training RL settings such as (single-turn) DPO, prior work, including VPO (Cen et al., 2024) and XPO (Xie et al., 2024), has shown that augmenting the standard negative log-likelihood objective with an additional reward-biasing term can effectively promote exploratory behavior during optimization. While promising, these methods remain confined to the (single-turn) DPO regime and thus do not establish the broader applicability of exploration-inducing bias across alternative RL algorithms. In addition, they rely solely on Chain-of-Thought decoding and therefore do not investigate how such techniques interact with modern test-time scaling strategies.

Motivated by these limitations, we generalize the idea of reward biasing to an iterative policy optimization framework and show that exploration-aware updates can be seamlessly incorporated into a wider family of RL methods, including both iterative DPO (iDPO) and GRPO. This demonstrates that exploration-enhancing principles are compatible with diverse policy optimization paradigms. Moreover, when combined with test-time scaling techniques, this integration consistently yields further performance gains.

Concretely, at iteration $t$, we encourage the reward function $r^{(t)}$ to maximize the RL objective $J^*(r^{(t)}) = \max_\pi J(r^{(t)}, \pi)$ as defined in Eq. 1. Since the rewards of observed responses are pinpointed by the ground-truth reward function or observed preferences, this biasing term effectively leads to an upper-confidence bound estimation of the rewards of unexplored responses and therefore promotes exploration. However, Eq. 1 also highlights a fundamental *reward-shift ambiguity*. For any prompt-dependent shift $c(x)$, two reward functions related by $r_1(x, y) = r_2(x, y) + c(x)$ produce identical pairwise differences: $r_1(x, y_w) - r_1(x, y_l) = r_2(x, y_w) - r_2(x, y_l)$, and therefore yield the same optimal solution. Yet, their maximized values differ by a constant offset: $J^\star(r_1) = J^\star(r_2) + \mathbb{E}_{x \sim \mathcal{D}}[c(x)]$, meaning the optimization target becomes ill-defined unless the shift is fixed. To remove this ambiguity, we impose a normalization constraint on $r^{(t)}$ by requiring its

expectation under the previous policy $\pi^{t-1}$ to be zero:

$$\mathcal{R}^t = \left\{ r^{(t)} \ \middle| \ \mathbb{E}_{x \sim \mathcal{D}, \, y \sim \pi^{t-1}(\cdot|x)} \left[ r^{(t)}(x, y) \right] = 0 \right\}, \tag{7}$$

Under this constraint, and leveraging the closed-form solution from DPO (Rafailov et al., 2023), the optimal value term $J^\star(r^{(t)})$ can be expressed as:

$$J^\star(r^{(t)}) = -\beta \, \mathbb{E}_{x \sim \mathcal{D}, y \sim \pi^{t-1}(\cdot|x)} \left[ \log \pi_{r^{(t)}}(y|x) - \log \pi_{\text{ref}}(y|x) \right], \tag{8}$$

where $\pi_{r^{(t)}}(y|x) = \frac{\pi_{\text{ref}}(y|x) \exp(r^{(t)}(x,y)/\beta)}{Z(r^{(t)}, x)}$ denotes the closed form optimal policy induced by reward function $r^{(t)}$, and $Z(r^{(t)}, x)$ denotes the corresponding partition function. A full derivation is provided in Appendix A.1. This closed-form representation transforms the maximization of the optimal value $J^\star(r^{(t)})$ into an equivalent optimization over the reward-parameterized policy $\pi_{r^{(t)}}$ at iteration $t$.

Consequently, the optimization of $J^\star(r^{(t)})$ can be reformulated directly in the policy space as:

$$\arg\max_{r^{(t)} \in \mathcal{R}^t} J^*(r^{(t)}) = \arg\max_{\pi_{r^{(t)}} : r^{(t)} \in \mathcal{R}^t} \left\{ -\beta \mathbb{E}_{x \sim \mathcal{D}, \, y \sim \pi^{t-1}} [\log \pi_{r^{(t)}}(y|x) - \log \pi_{\text{ref}}(y|x)] \right\}$$

$$= \arg\max_{\pi_\theta} \left\{ -\beta \mathbb{E}_{x \sim \mathcal{D}, \, y \sim \pi^{t-1}} [\log \pi_\theta(y|x) - \log \pi_{\text{ref}}(y|x)] \right\} \tag{9}$$

where in the second step we remove the constraint $r^{(t)} \in \mathcal{R}^t$ since, for any policy $\pi_\theta$, one can construct a reward function $r^{(t)} \in \mathcal{R}^t$ such that $\pi_\theta = \pi_{r^{(t)}}$.

Observe that the term $\log \pi_{\text{ref}}(y|x)$ does not depend on $\pi_\theta$ and thus does not influence the maximizer. Replacing it with $\log \pi^{t-1}(y|x)$ yields an equivalent objective: $-\beta \mathbb{E}_{x \sim \mathcal{D}, \, y \sim \pi^{t-1}} [\log \pi_\theta(y|x) - \log \pi^{t-1}(y|x)] = -\beta \, \mathbb{D}_{\text{KL}}(\pi^{t-1} \| \pi_\theta)$. This expression makes explicit that maximizing $J^*(r^{(t)})$ encourages the current policy $\pi_\theta$ to diverge from the previous iterate $\pi^{t-1}$, forming the foundation of our exploration-driven optimization strategy.

In the following sections, we discuss how this reward biasing mechanism can be instantiated in both the iDPO and GRPO settings, enabling the policy to explore a broader solution space during iterative optimization.

**Exploration-Driven Iterative Direct Preference Optimization.** In Iterative Direct Preference Optimization, incorporating an additional reward-bias term leads to the modified objective:

$$\mathcal{L}_{\text{ED-iDPO}} = \mathcal{L}_{\text{iDPO}} - \alpha J^*(r^{(t)}), \tag{10}$$

where $\alpha > 0$ controls the influence of the bias.

The corresponding optimal policy at iteration $t$ is therefore given by

$$\pi_{\text{ED-iDPO}}^t = \arg\min_{\pi_\theta} \left\{ -\mathbb{E}_{(x, y_w, y_l) \sim \mathcal{D}} \left[ \log \sigma \left( \beta \log \frac{\pi_\theta(y_w \mid x)}{\pi_{\text{ref}}(y_w \mid x)} - \beta \log \frac{\pi_\theta(y_l \mid x)}{\pi_{\text{ref}}(y_l \mid x)} \right) \right] \right.$$
$$\left. + \alpha \beta \mathbb{E}_{x \sim \mathcal{D}, y \sim \pi^{t-1}(\cdot|x)} \left[ \log \pi_\theta(y|x) - \log \pi^{t-1}(y|x) \right] \right\}, \tag{11}$$

The second term in Eq. 11 is equivalent to $-\mathbb{D}_{\text{KL}}(\pi^{t-1} \| \pi_\theta)$, which encourages the updated policy $\pi_\theta$ to *move away* from the previous iterate $\pi^{t-1}$, thereby inducing exploration across iterations. In contrast, the original KL term in DPO loss (Eq. 4), $\mathbb{D}_{\text{KL}}(\pi_\theta \| \pi_{\text{ref}})$ anchors the policy within a **trust region** centered on the reference model $\pi_{\text{ref}}$. Together, these two complementary biasing terms maintain a principled balance: the algorithm is encouraged to explore beyond the previous iterate while remaining sufficiently close to the reference policy to avoid instability or model collapse (Shumailov et al., 2024). This interpretation also clarifies why `EDO` is not equivalent to merely increasing the standard KL regularization coefficient. A larger forward KL penalty mainly shrinks updates toward $\pi_{\text{ref}}$, whereas the reverse-KL term $-\mathbb{D}_{\text{KL}}(\pi^{t-1} \| \pi_\theta)$ changes the update direction by explicitly discouraging repeated concentration on the previous iterate's dominant modes.

**Exploration-Driven Group Relative Policy Optimization.** To demonstrate the generalizability of `EDO`, we show that it can also be seamlessly integrated with GRPO (Shao et al., 2024), resulting in `ED-GRPO`. It leverages GRPO's strength in enhancing the reasoning ability of policy models, while also smoothing the output probability distribution to promote exploration of more diverse solutions.

To express $J^*(r)$ in a closed-form solution as before, we rewrite the GRPO objective (Eq. 6) into the standard RL optimization form (Eq. 1). The derivation below should be interpreted as a motivating approximation: for analytic clarity, we omit clipping and replace the finite sampled group objective by its expectation under a sufficiently large group size. Actual training still uses the finite-group, clipped GRPO objective; the theory is intended to explain the induced update direction rather than claim exact equality for every finite-sample implementation. Under these approximations, Eq. 6 can be written as:

$$
\begin{aligned}
J^{\mathrm{GRPO}} &= \mathbb{E}_{x\sim\mathcal{D},\, y_t\sim\pi_{\mathrm{old}}(\cdot|x,y_{<t})}\left[\frac{1}{|y|}\sum_{t=1}^{|y|}\frac{\pi_\theta(y_t\mid x,y_{<t})}{\pi_{\mathrm{old}}(y_t\mid x,y_{<t})}\hat{A}_t \;-\; \beta\,\mathbb{D}_{\mathrm{KL}}\big(\pi_\theta\,\|\,\pi_{\mathrm{ref}}\big)\right] \\
&= \mathbb{E}_{x\sim\mathcal{D},\, y_t\sim\pi_\theta(\cdot|x,y_{<t})}\Bigg[\underbrace{\frac{1}{|y|}\sum_{t=1}^{|y|}\hat{A}_t}_{J_{\mathrm{R}}:\,\text{Expected Cumulative Reward}} \quad -\;\beta\underbrace{\sum_{t=1}^{|y|}\log\frac{\pi_\theta(y_t\mid x,y_{<t})}{\pi_{\mathrm{ref}}(y_t\mid x,y_{<t})}}_{J_{\mathrm{KL}}:\,\text{KL Penalty}}\Bigg]
\end{aligned}
\tag{12}
$$

where the GRPO advantage function is defined using outcome supervision as $\hat{A}_t = \hat{r}(x,y) = \frac{r(x,y)-\mu(x)}{\sigma(x)}$ for $t \in [1:|y|]$. We use the shorthand notations $\mu(x) = \mathrm{mean}_{y'\sim\pi_{\mathrm{old}}(\cdot|x)}\,[r(x,y')]$ and $\sigma(x) = \mathrm{std}_{y'\sim\pi_{\mathrm{old}}(\cdot|x)}\,[r(x,y')]$ to denote the mean and standard deviation of the group-relative rewards, respectively. The second line of Eq. 12 follows directly from applying importance sampling.

Since the per-token advantage is constant across positions for a given sequence, the expected cumulative reward under $\pi_{\mathrm{old}}$ can be represented as:

$$
\begin{aligned}
\mathbb{E}_{x\sim\mathcal{D},\, y_t\sim\pi_{\mathrm{old}}(\cdot|x,y_{<t})}\left[\frac{1}{|y|}\sum_{t=1}^{|y|}\hat{A}_t\right] &= \mathbb{E}_{x\sim\mathcal{D},\, y_t\sim\pi_{\mathrm{old}}(\cdot|x,y_{<t})}\big[\hat{r}(x,y)\big] \\
&= \mathbb{E}_{x\sim\mathcal{D},\, y\sim\pi_{\mathrm{old}}(\cdot|x)}\left[\frac{r(x,y)-\mathrm{mean}_{y'\sim\pi_{\mathrm{old}}(\cdot|x)}\,[r(x,y')]}{\mathrm{std}_{y'\sim\pi_{\mathrm{old}}(\cdot|x)}\,[r(x,y')]}\right] \\
&= 0
\end{aligned}
\tag{13}
$$

Leveraging the closed-form relationship between the reward and its optimal policy in token-level MDP (Appendix A.2), we can express $J^*(r)$ as:

$$
J^*(r) = -\beta\,\mathbb{E}_{x\sim\mathcal{D},\, y_t\sim\pi_{\mathrm{old}}(\cdot|x,y_{<t})}\left[\sum_{t=1}^{|y|}\log\frac{\pi_r(y_t\mid x,y_{<t})}{\pi_{\mathrm{ref}}(y_t\mid x,y_{<t})}\right]
\tag{14}
$$

Finally, substituting this expression into the reward-biased objective yields the closed-form characterization of the optimal policy under the `ED-GRPO` update:

$$
\begin{aligned}
\pi_{\mathrm{ED-GRPO}} = \arg\min_{\pi_\theta}\Bigg\{&\mathbb{E}_{x\sim\mathcal{D},\, \{y_i\}_{i=1}^G\sim\pi_{\mathrm{old}}(\cdot|x)}\Bigg[\frac{1}{G}\sum_{i=1}^G\frac{1}{|y_i|}\sum_{t=1}^{|y_i|}\Big(\min\Big(\frac{\pi_\theta(y_{i,t}|x,y_{i,<t})}{\pi_{\mathrm{old}}(y_{i,t}|x,y_{i,<t})}\hat{A}_{i,t}, \\
&\mathrm{clip}\Big(\frac{\pi_\theta(y_{i,t}|x,y_{i,<t})}{\pi_{\mathrm{old}}(y_{i,t}|x,y_{i,<t})},1-\epsilon,\,1+\epsilon\Big)\hat{A}_{i,t}\Big) - \beta\,\mathbb{D}_{\mathrm{KL}}\big(\pi_\theta\,\|\,\pi_{\mathrm{ref}}\big) + \alpha\beta\log\frac{\pi_\theta(y_{i,t}|x,y_{i,<t})}{\pi_{\mathrm{ref}}(y_{i,t}|x,y_{i,<t})}\Big)\Bigg]\Bigg\},
\end{aligned}
\tag{15}
$$

Consistent with the iDPO formulation, the last term in Eq. 15 can be rewritten as $\log\frac{\pi_\theta(y_{i,t}|x,y_{i,<t})}{\pi_{\mathrm{old}}(y_{i,t}|x,y_{i,<t})}$, which corresponds to the token-level KL divergence $\mathbb{D}_{\mathrm{KL}}(\pi_{\mathrm{old}}\,\|\,\pi_\theta)$ and still yields the same optimal policy. Intuitively, this biased objective also encourages the current policy $\pi_\theta$ to diverge from the previous policy $\pi_{\mathrm{old}}$, while still constrained within the **trust region** around $\pi_{\mathrm{ref}}$ to prevent abrupt model collapse.

## 4.3 Overall Pipeline

Our proposed `EDO` is an iterative online training method that starts with an SFT fine-tuned model $\pi_{\mathrm{SFT}}$. In each iteration, newly generated problem-response pairs are incorporated into the training of the policy model.

During inference, `EDO` is further combined with test-time computation techniques such as self-consistency and best-of-N. All prompts used for data generation and evaluation are included in Appendix G.

**Collecting New Data**    Given the current policy model $\pi_t$, we generate $N$ candidate solutions $\mathcal{Y}_t = \{y_i^t\}_{i=1}^N$ for each question. The reward for each solution is computed based on its correctness, using a simple rule-based metric in our experiments, i.e., $r_i^t = 1$ if $y_i^t = y$, and 0 otherwise, where $y$ is the ground-truth answer provided in the training dataset. Depending on the specific optimization objective, data collection can be categorized into the following two settings:

1. **Data Collection for `ED-iDPO`.** For preference-based optimization, we construct a paired dataset $D_t^{pair}$ using the generated outputs $\mathcal{Y}_t$ from the current policy model $\pi_t$. The paired data satisfies the preferred (chosen) responses have higher rewards than the non-preferred (rejected) ones. In the binary reward case, we devide the generated responses into two sets:

$$\mathcal{Y}_t^w = \left\{ y_i^t \mid r_i^t = 1 \right\}, \quad \mathcal{Y}_t^l = \left\{ y_i^t \mid r_i^t = 0 \right\}. \tag{16}$$

We then construct $S$ preference pairs $(y_i^w, y_i^l)$ by iterating over both sets:

$$D_t^{\text{pair}} = \left\{ \left( y_i^{w_s}, y_i^{l_s} \right) \mid x_i \in D \text{ and } s \in [S] \right\}. \tag{17}$$

2. **Data Collection for `ED-GRPO`.** When GRPO is used as the optimization strategy, the output group $\mathcal{Y}_t$ directly provides group-level feedback for updating the policy according to Eq. 15.

**Iterative Self-Improvement**    Our iterative training procedure involves learning a sequence of models $\pi_\theta^0, \ldots, \pi_\theta^T$, where $\pi_\theta^0$ can be initialized from $\pi_{\text{SFT}}$ and each successive model is trained using either the preference pairs $D_t^{\text{pair}}$ or the generated outputs $\mathcal{Y}_t$ collected from the current model $\pi_\theta^t$. By incorporating **Exploration-Driven Optimization** introduced in Section 4.2, the complete training pipeline is summarized in Algorithm 1.

---

**Algorithm 1 Exploration-Driven** Training Pipeline

---

**Input:** $\mathcal{X}$: prompt set; $\mathcal{Y}$: ground truth solution set; $\pi_\theta^0$: initialized policy; $T$: max iteration epoch
**Output:** $\pi_\theta^T$: Exploration-Driven Policy
**for** $t$ in range($T$) **do**
    Generate $\mathcal{Y}_t = \{y_i^t\}_{i=1}^N$ for each question $x \in \mathcal{X}$, where $y_i^t \sim \pi_\theta^t(\cdot \mid x)$
    Collect preference dataset $D_t^{\text{pair}}$ as Eq. 17 (**iDPO**) or directly use $\mathcal{Y}_t$ (**GRPO**)
    Train $\pi_\theta^t$ with `ED-iDPO` (Eq. 11) or `ED-GRPO` (Eq. 15) to get $\pi_\theta^{t+1}$
**end for**
**return** $\pi_\theta^T$

---

# 5 Experiments

We present comprehensive experiments across a diverse set of math reasoning tasks to demonstrate the enhanced capabilities of `EDO`, especially when combining with test-time computing.

## 5.1 Experimental Setup

**Tasks and Datasets.**    We consider the following three tasks for training in our experiments to evaluate the model's performance across varying levels of problem difficulty: (1) **GSM8K** (Cobbe et al., 2021), which consists of grade school math word problems; (2) **Math** (Hendrycks et al., 2021), which features high school math competition problems; and (3) **s1K** (Muennighoff et al., 2025), comprising the most challenging competition-level questions. For **s1K**, we use the 1,000-question version for efficiency and evaluate on **s1K_eval**, a separate set of 1000 questions drawn from the same sources but containing entirely different problems. To further assess out-of-distribution (OOD) generalization, we include additional evaluation on the following datasets: **AIME24**, **AIME25**, **MATH500** (Hendrycks et al., 2021), **Minerva Math** (Lewkowycz et al., 2022) and **Olympiad Bench** (He et al., 2024). Dataset details are available in Appendix C.

Table 1: In-distribution performance across multiple reasoning benchmarks. For **s1K**, we use the supervised fine-tuned `Qwen2.5-7B-Instruct` and `LLaMA3.1-8B-Instruct` models as our base models. For **Math** and **GSM8K**, we use the vanilla `Qwen2.5-7B-Instruct` and `LLaMA3-8B-Instruct` models as the respective bases. The **s1K** results are reported on the curated **s1K$_{eval}$** split. We highlight the best score in **bold** and the second-best with an underline. All methods employ both Chain-of-Thought (Wei et al., 2022) and Self-Consistency (Wang et al., 2022) during inference.

| Dataset ($\rightarrow$) | s1K (Qwen2.5-7B-Instruct-SFT) | | s1K (LlaMA3.1-8B-Instruct-SFT) | | Math (Qwen2.5-7B-Instruct) | | GSM8K (LlaMA3-8B-Instruct) | |
|---|---|---|---|---|---|---|---|---|
| Method ($\downarrow$) / Metrics ($\rightarrow$) | Acc. (%) | $\Delta$ (%) | Acc. (%) | $\Delta$ (%) | Acc. (%) | $\Delta$ (%) | Acc. (%) | $\Delta$ (%) |
| **CoT** | 34.6 | - | 17.2 | - | 72.8 | - | 79.3 | - |
| *+ Self-Consistency* | 46.1 | +11.5 | 27.0 | +9.8 | 76.2 | +3.4 | 87.4 | +8.1 |
| **ETO** | 37.0 | +2.4 | 17.7 | +0.5 | 73.2 | +0.4 | 81.5 | +2.2 |
| *+ Self-Consistency* | 47.3 | +12.7 | 26.7 | +9.5 | 77.4 | +4.6 | 89.2 | +9.9 |
| **GRPO** | 35.1 | +0.5 | 18.9 | +1.7 | 74.2 | +1.4 | 83.6 | +4.3 |
| *+ Self-Consistency* | 46.7 | +12.1 | 27.3 | +10.1 | 78.4 | +5.6 | 88.7 | +9.4 |
| **DAPO** | 36.1 | +1.5 | 18.3 | +1.1 | 75.2 | +2.4 | 82.1 | +2.8 |
| *+ Self-Consistency* | 48.1 | +13.5 | 26.7 | +9.5 | 78.4 | +5.6 | 88.0 | +8.7 |
| **ED-iDPO** | 36.9 | +2.3 | 18.7 | +1.5 | 73 | +0.2 | 81.5 | +2.2 |
| *+ Self-Consistency* | **48.9** | **+14.3** | **28.6** | **+11.4** | 78.6 | +5.8 | 88.9 | +9.6 |
| **ED-GRPO** | 38.6 | +4.0 | 21.0 | +3.8 | 76.4 | +3.6 | 86.3 | +7.0 |
| *+ Self-Consistency* | 48.4 | +13.8 | 28.1 | +10.9 | **79.4** | **+6.6** | **90.4** | **+11.1** |

**Baselines.** We consider the following baselines: (1) *Reasoning-based baselines*, including Chain-of-Thought (CoT) (Wei et al., 2022) and Self-Consistency (Wang et al., 2022) (2) *Training-based baselines*, we mainly compare with ETO (Song et al., 2024) (a self-improvement variant of online DPO), GRPO (Shao et al., 2024), DAPO (Yu et al., 2025) and further combine them with test-time scaling method like Self-Consistency. Baseline details can be found in Appendix D.

**Evaluation metrics.** For **Math** and **GSM8K**, we compute *accuracy* via exact matching with rule-based reference answers. For more challenging benchmarks such as **s1K$_{eval}$**, exact matching becomes unreliable due to the open-form nature of the answers. Following prior work (Muennighoff et al., 2025), we evaluate accuracy using an LLM-as-a-judge protocol[1], employing `gemini-2.5-flash-lite` as the judging model. This judge-based evaluation is applied consistently across all s1K-related evaluation datasets, including AIME24, AIME25, MATH500, Minerva Math, Olympiad Bench, and s1K$_{eval}$.

**Implementations Details.** For experiments on **s1K**, we utilize `Qwen2.5-7B-Instruct` and `LLaMA3.1-8B-Instruct` as the backbone language models. For **GSM8K** and **Math**, the backbone model are `LLaMA3-8B-Instruct` and `Qwen2.5-7B-Instruct`, respectively. Given that **s1K** involves extremely long sequence generation (exceeding 10,000 tokens), we perform one epoch of supervised fine-tuning (SFT) on the provided dataset before applying RL post-training. In contrast, we **skip** the SFT stage for the **GSM8K** and **Math** tasks. During evaluation, we report both greedy decoding results and results obtained with Self-Consistency. For the Self-Consistency setting, we generate 10 responses per question using temperature $t = 1.0$, and determine the final prediction via majority voting. All reported Self-Consistency results are averaged over 3 independent rollouts. Additional implementation details are provided in Appendix E.

## 5.2 Main Results

**In-distribution Performance.** Table 1 summarizes the in-distribution performance across three reasoning datasets. Under greedy decoding, `ED-iDPO` and `ED-GRPO` yield average improvements of 1.5% and 4.6%, respectively, over standard Chain-of-Thought prompting. When combined with self-consistency, the gains increase substantially to 10.1% and 10.6%, and both methods further outperform self-consistency alone by 1.9% and 2.4% on average. These results indicate that the proposed methods more effectively navigate the solution space and identify correct outputs.

Moreover, `ED-iDPO` consistently surpasses ETO across both greedy and self-consistency settings, while `ED-GRPO` outperforms GRPO and its variant DAPO by 2.6% and 2.7% under greedy decoding and by 1.4%

---

[1]https://github.com/simplescaling/s1

Table 2: Out-of-distribution evaluation results on five math reasoning datasets. We use the supervised fine-tuned `Qwen2.5-7B-Instruct` and `LLaMA3.1-8B-Instruct` models on **s1K** as our base models. For all datasets, we employ the same prompt as in **s1k**.

| | | | | | | | | | | | | |
|---|---|---|---|---|---|---|---|---|---|---|---|---|
| *Qwen2.5-7B-Instruct-SFT* | | | | | | | | | | | | |
| Dataset (→) | AIME24 | | AIME25 | | MATH500 | | Minerva Math | | Olympiad Bench | | Overall | |
| Method (↓) / Metrics (→) | Acc. (%) | Δ (%) | Acc. (%) | Δ (%) | Acc. (%) | Δ (%) | Acc. (%) | Δ (%) | Acc. (%) | Δ (%) | Acc. (%) | Δ (%) |
| **CoT** | 13.3 | - | 6.7 | - | 74.6 | - | 36.4 | - | 40.4 | - | 34.3 | - |
| *+ Self-Consistency* | 16.7 | +3.4 | 13.3 | +6.6 | 84.4 | +9.8 | 41.9 | +5.5 | 52.1 | +11.7 | 41.7 | +7.4 |
| **ETO** | 16.7 | +3.4 | 10.0 | +3.3 | 75.4 | +0.8 | 37.9 | +1.5 | 42.7 | +2.3 | 36.5 | +2.3 |
| *+ Self-Consistency* | **23.3** | **+10.0** | **20.0** | **+13.3** | 84.8 | +10.2 | 42.3 | +5.9 | 51.0 | +10.6 | 44.3 | +10.0 |
| **GRPO** | 16.7 | +3.4 | 13.3 | +6.6 | 76.2 | +1.6 | 39.7 | +3.3 | 42.9 | +2.5 | 37.8 | +3.5 |
| *+ Self-Consistency* | 20.0 | +6.7 | 16.7 | +10.0 | 83.8 | +9.2 | 44.1 | +7.7 | 52.4 | +12.0 | 43.4 | +9.1 |
| **DAPO** | 16.7 | +3.4 | 10.0 | +3.3 | 75.8 | +1.2 | 38.2 | +1.8 | 41.8 | +1.4 | 36.5 | +2.2 |
| *+ Self-Consistency* | 16.7 | +3.4 | 16.7 | +10.0 | 84.8 | +10.2 | 45.2 | +8.8 | 51.6 | +11.2 | 43.0 | +8.7 |
| **ED-iDPO** | 16.7 | +3.4 | 13.3 | +6.6 | 77.0 | +2.4 | 38.6 | +2.2 | 42.6 | +2.2 | 37.6 | +3.4 |
| *+ Self-Consistency* | 23.3 | +10.0 | 20.0 | +13.3 | 84.6 | +10.0 | 44.5 | +8.1 | **52.8** | **+12.4** | 45.0 | +10.8 |
| **ED-GRPO** | 23.3 | +10.0 | 16.7 | +10.0 | 77.6 | +3.0 | 41.2 | +4.8 | 43.9 | +3.5 | 40.5 | +6.3 |
| *+ Self-Consistency* | **23.3** | **+10.0** | **20.0** | **+13.3** | **85.0** | **+10.4** | **48.2** | **+11.8** | 52.5 | +12.1 | **45.8** | **+11.5** |
| *LLaMA3.1-8B-Instruct-SFT* | | | | | | | | | | | | |
| Dataset (→) | AIME24 | | AIME25 | | MATH500 | | Minerva Math | | Olympiad Bench | | Overall | |
| Method (↓) / Metrics (→) | Acc. (%) | Δ (%) | Acc. (%) | Δ (%) | Acc. (%) | Δ (%) | Acc. (%) | Δ (%) | Acc. (%) | Δ (%) | Acc. (%) | Δ (%) |
| **CoT** | 0.0 | - | 0.0 | - | 45.6 | - | 18.4 | - | 18.7 | - | 16.5 | - |
| *+ Self-Consistency* | 3.3 | +3.3 | 3.3 | +3.3 | 57.0 | +11.4 | 22.8 | +4.4 | 28.2 | +9.5 | 22.9 | +6.4 |
| **ETO** | 3.3 | +3.3 | 3.3 | +3.3 | 46.0 | +0.4 | 19.1 | +0.7 | 19.1 | +0.4 | 18.2 | +1.6 |
| *+ Self-Consistency* | **6.7** | **+6.7** | 3.3 | +3.3 | 61.6 | +16.0 | 24.3 | +5.9 | 27.7 | +9.0 | 24.7 | +8.2 |
| **GRPO** | 3.3 | +3.3 | 0.0 | - | 45.4 | -0.2 | 21.7 | +3.3 | 19.1 | +0.4 | 17.9 | +1.4 |
| *+ Self-Consistency* | 3.3 | +3.3 | 0.0 | - | 57.2 | +11.6 | **27.2** | **+8.8** | 27.9 | +9.2 | 23.1 | +6.6 |
| **DAPO** | 3.3 | +3.3 | 0.0 | - | 44.4 | -1.2 | 21.7 | +3.3 | 18.4 | -0.3 | 17.6 | +1.0 |
| *+ Self-Consistency* | 3.3 | +3.3 | 0.0 | - | 54.4 | +8.8 | 25.4 | +7.0 | 26.1 | +7.4 | 21.8 | +5.3 |
| **ED-iDPO** | **6.7** | **+6.7** | 3.3 | +3.3 | 47.0 | +1.4 | 23.2 | +4.8 | 19.6 | +0.9 | 20.0 | +3.4 |
| *+ Self-Consistency* | **6.7** | **+6.7** | 3.3 | +3.3 | **62.0** | **+16.4** | 26.8 | +8.4 | **30.1** | **+11.4** | **25.8** | **+9.2** |
| **ED-GRPO** | 3.3 | +3.3 | 6.7 | +6.7 | 49.2 | +3.6 | 23.2 | +4.8 | 20.6 | +1.9 | 20.6 | +4.1 |
| *+ Self-Consistency* | 3.3 | +3.3 | 6.7 | +6.7 | 61.4 | +15.8 | 26.5 | +8.1 | 29.5 | +10.8 | 25.5 | +8.9 |

and 1.5% under majority voting. This consistent superiority demonstrates the strong generalization capability of `EDO` across diverse RL training paradigms. We attribute these gains to the synergy between `EDO`, which expands the model's exploration capacity, and self-consistency, which implicitly functions as a Q-value estimator to select high-quality responses.

**Plug-and-Play: Out-of-distribution Performance.** The tuned policy can be seamlessly applied to out-of-distribution datasets in a plug-and-play manner, eliminating the need for retraining or additional modifications. An exploration-enhanced policy trained on **s1K** also exhibits strong cross-dataset generalization by enabling broader exploration of the solution space in the target domains. Table 2 presents the out-of-distribution results for `ED-iDPO` and `ED-GRPO`, using both supervised fine-tuned `Qwen-2.5-7B-Instruct` and `LLaMA3.1-8B-Instruct` as base models. Relative to their non-exploration baselines (ETO and GRPO), our methods yield average gains of **0.8%** and **2.4%** on Qwen and **0.8%** and **2.3%** for LLaMA across all five datasets. These results demonstrate that enhancing exploration not only improves transferability across heterogeneous data distributions but also provides broad applicability across different policy models.

## 5.3 Ablation Studies

### 5.3.1 Effectiveness of the Iterative Framework

In this section, we evaluate the effectiveness of the proposed iterative training framework on Math (Hendrycks et al., 2021) and GSM8K (Cobbe et al., 2021), while adopting Self-Consistency as the test-time scaling strategy. As shown in Fig. 2, our exploration-driven optimization approach consistently improves performance

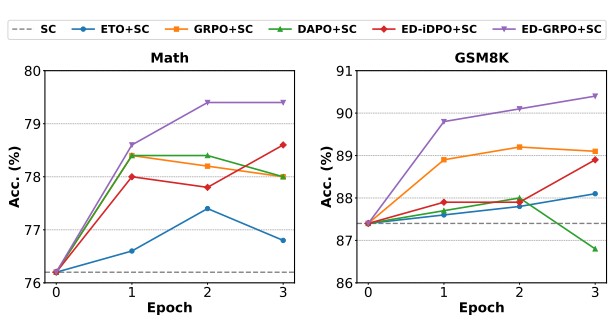
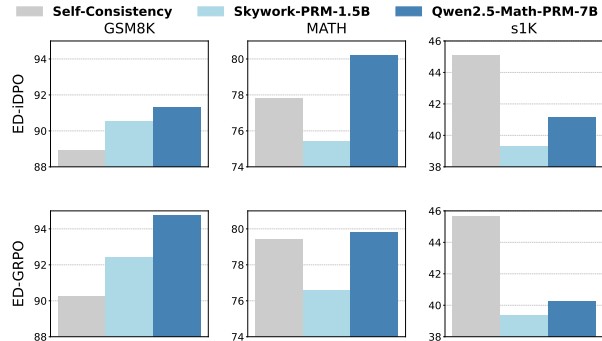

Figure 2: Accuracy vs. Epochs on Math and GSM8K across different optimization methods. Self-Consistency (SC) is applied at inference, and results are averaged over three runs.

Figure 3: Test-time scaling performance of `EDO` on GSM8K, MATH and s1K datasets. `Skywork-PRM-1.5B` and `Qwen2.5-Math-PRM-7B` represent RMs with BoN.

and training stability for both variants, `ED-iDPO` and `ED-GRPO`. Notably, both methods demonstrate enhanced robustness to over-optimization (Gao et al., 2023). For example, while ETO exhibits a 0.8% performance degradation after the second training epoch on Math, `ED-iDPO` maintains steady gains across three iterations. A similar trend is observed for GRPO, whose performance remains flat and even declines slightly after the first epoch on both Math and GSM8K, whereas `ED-GRPO` preserves stable performance throughout training. These results highlight the effectiveness of our proposed objectives in Eq. 11 and Eq. 15, which promote diverse solution sampling and mitigate the risk of over-optimizing toward suboptimal modes.

### 5.3.2 Analysis of Test-Time Scaling Strategies

Our method integrates seamlessly with a range of test-time scaling strategies, including Self-Consistency (Wang et al., 2022), Best-of-N (BoN), and the kernel-based tree-search procedure described in Appendix I. For BoN evaluation, we primarily adopt `Qwen2.5-Math-PRM-7B` (Zhang et al., 2025b) and `Skywork-PRM-1.5B` (Skywork o1 Team, 2024) as reward models (RMs), selecting the candidate whose final reasoning step receives the highest RM-assigned reward. Generation configurations follow Section 5.2, with additional RM specifications provided in Appendix E.4. For Math and GSM8K, we report results on their respective evaluation splits. For s1K, we report the average performance over six benchmarks: AIME24, AIME25, MATH500, Minerva Math, OlympiadBench, and s1K$_{\mathrm{eval}}$. As illustrated in Figure. 3, BoN effectively identifies high-quality solutions for relatively easy tasks such as GSM8K, consistently outperforming majority voting. For medium-difficulty tasks such as Math, Self-Consistency begins to surpass BoN when paired with a weaker RM (i.e., `Skywork-PRM-1.5B`), although it still underperforms compared to the strongest BoN baseline. In contrast, for challenging datasets like s1K, Self-Consistency achieves the best performance, highlighting the difficulty of transferring RMs to complex reasoning tasks. Across all settings, `Qwen2.5-Math-PRM-7B` consistently outperforms `Skywork-PRM-1.5B`, demonstrating the importance of high-quality RMs for robust solution evaluation. A comprehensive comparison of test-time scaling strategies across benchmarks, including results from the tree-based search method, is presented in Appendix F.1.

### 5.3.3 Effectiveness of Exploration

In this section, we investigate the role of exploration in our framework from two complementary perspectives: training dynamics and sequence-level output diversity. We first analyze how `EDO` preserves exploration during optimization. Since `EDO` explicitly encourages the policy to deviate from the old policy while remaining within the trust region induced by the reference policy, it is expected to counteract the well-known entropy collapse commonly observed in RL-based optimization.

To validate this, we conduct controlled comparisons on GSM8K and Math. As shown in Figure 4, the standard GRPO objective rapidly drives the model toward low-entropy, low-diversity behavior as training progresses. DAPO partially alleviates this issue by asymmetrically modifying its clipping range, which slows

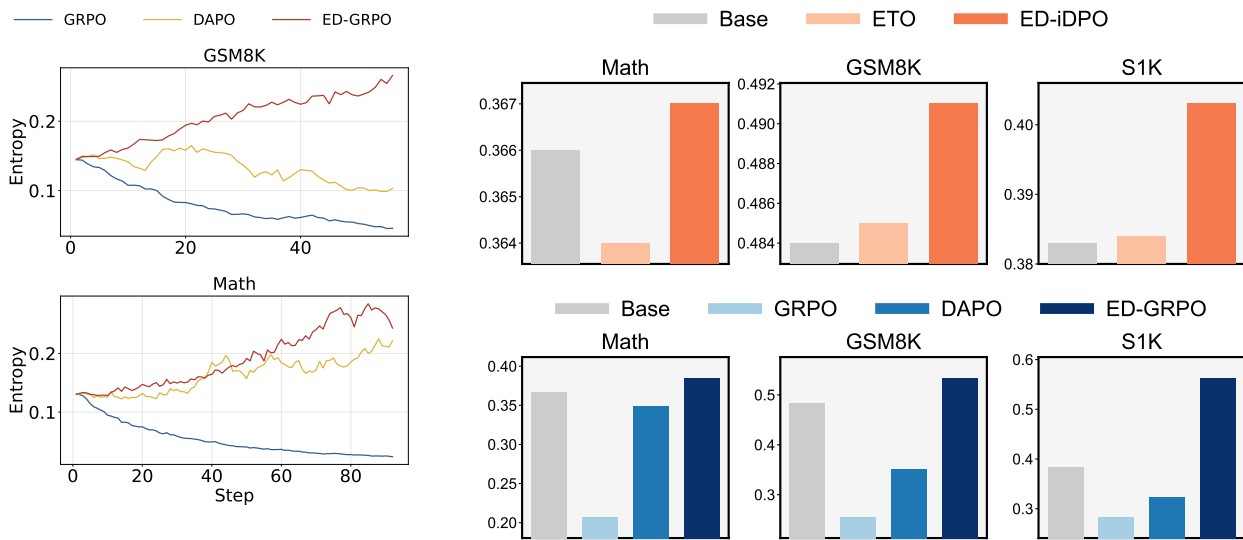

Figure 4: Evolution of policy entropy during training on GSM8K and Math datasets.

Figure 5: Distinct-4 diversity of generated responses on Math, GSM8K, and S1K. Higher values indicate more diverse outputs.

the entropy decay on Math; however, it still undergoes a clear downward trend on GSM8K. In contrast, our exploration-augmented `ED-GRPO` consistently prevents entropy collapse on both datasets, maintaining significantly higher entropy throughout training. Combined with the accuracy improvements reported in Table 1 and Table 2, these results demonstrate that the exploration term not only stabilizes exploration but also supports better generalization and reasoning performance.

We further evaluate the effect of exploration on sequence-level diversity. Following the evaluation setup in Section 5.2, we measure n-gram diversity (Li et al., 2015) (detailed in Appendix F.2) across Math, GSM8K, and S1K. As shown in Figure 5, standard training methods, including ETO, GRPO, and DAPO, substantially reduce response diversity after training. In contrast, both `ED-iDPO` and `ED-GRPO` reliably yield more diverse outputs, improving Distinct-4 by 5.2% and 47.0% on S1K, respectively. These findings indicate that our exploration term is not only effective at preventing entropy collapse during optimization, but also translates into richer and more varied model generations.

## 5.4 Case Study

Figure 6 provides a detailed case study on the MATH dataset, illustrating how exploration fundamentally changes the model's behavior. For this geometry problem, the vanilla iterative DPO-trained model generates three nearly identical solutions: all of them begin by "Constructing the Voronoi diagram" or "Understanding the geometry," and each proceeds through an almost identical sequence of steps before mistakenly concluding that the answer is 1/4. Despite minor surface-level differences, the underlying reasoning paths are essentially the same, resulting in three homogeneous trajectories that repeat the same flawed logic. As a result, both Self-Consistency and Best-of-N fail, since aggregating or ranking these near-duplicates yields the same incorrect answer.

In contrast, the `ED-iDPO` model produces a much more diverse set of reasoning paths. One solution starts by "defining coordinates and simplifying inequalities," another focuses on "identifying the region closer to the center," and a third analyzes the geometry by "visualizing bisectors and computing areas." These trajectories differ not only in wording but also in the actual solution strategies they employ, ranging from coordinate geometry to region decomposition to geometric symmetry arguments. This diversity leads to multiple distinct candidate answers, including the correct 1/2. As a result, simple test-time scaling strategies such as majority voting or Best-of-N can successfully amplify the correct answer.

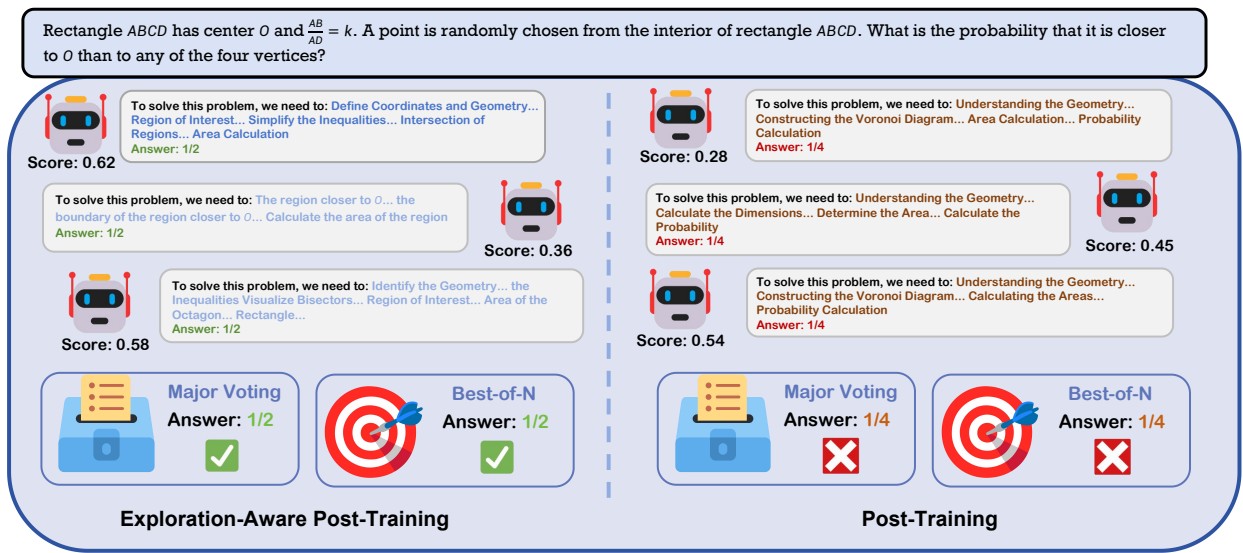

Figure 6: Case study of `EDO` on the MATH dataset. For the given question, the iterative online DPO-trained model consistently produces homogeneous solutions, leading to incorrect answers even when combined with test-time scaling methods. In contrast, the model trained with `ED-iDPO` generates significantly more diverse solutions, ultimately yielding the correct answer. For visualization purposes, we display only 3 candidate solutions.

This example highlights the core advantage of ED-iDPO: by encouraging a broader exploration of solution strategies, the model avoids collapsing onto a single incorrect mode, dramatically increasing the likelihood that test-time methods recover the correct answer.

# 6 Conclusion

We present `EDO`, a novel exploration-driven optimization strategy specifically designed to promote greater diversity in the solution space explored by large language models during inference. By incorporating an additional reward-biasing term, `EDO` effectively integrates with established reinforcement learning paradigms, yielding enhanced variants `ED-iDPO` and `ED-GRPO`. Our extensive empirical evaluation demonstrates significant improvements in both solution diversity and reasoning performance, particularly notable in challenging out-of-distribution scenarios. Furthermore, `EDO` produces more diverse generations and remains stable under continued optimization, highlighting its generality and practical effectiveness. We conduct extensive experiments across different models and settings to validate our method. We hope our work provides key insights for the LLM and RL community.

### Acknowledgments

We thank the reviewers and action editors for their valuable feedback. This work was supported in part by the ONR under grant N000142512173, and by the NSF under grants ECCS: 2401391 and IIS: 2403240, and Dolby support.

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

# A Derivation for `EDO`

## A.1 Derivation details for $J^*(r)$

In iterative RL optimization setup, the KL-regularized objective function at the $t$-th iteration can be formulated as:

$$J^*(r^{(t)}) = \max_{\pi_\theta} J(r^{(t)}, \pi_\theta) = \max_{\pi_\theta} \mathbb{E}_{x\sim\mathcal{D}, y\sim\pi_\theta(\cdot|x)} \left[ r^{(t)}(x,y) - \beta\, \mathbb{D}_{\mathrm{KL}} \left( \pi_\theta(y \mid x) \,\|\, \pi_{\mathrm{ref}}(y \mid x) \right) \right], \quad (18)$$

The objective admits a closed-form optimal policy as shown in DPO (Rafailov et al., 2023), given by:

$$\forall (x,y) \in \mathcal{X} \times \mathcal{Y}: \quad \pi_r^{(t)}(y \mid x) = \frac{\pi_{\mathrm{ref}}(y \mid x) \exp(r^{(t)}(x,y)/\beta)}{Z(x)}, \quad (19)$$

where the normalization term $Z(x) = \sum_{y'\in\mathcal{Y}} \pi_{\mathrm{ref}}(y'|x) \exp(r^{(t)}(x,y')/\beta)$ is also known as the partition function.

By utilizing the closed-form policy in Eq. 19, we can derive a simplified expression for $J^\star(r^{(t)})$ as follows:

$$\begin{aligned}
J^\star(r^{(t)}) &= \mathbb{E}_{x\sim\mathcal{D}, y\sim\pi_r^{(t)}(\cdot|x)} \left[ r^{(t)}(x,y) - \beta\left( \log\pi_r^{(t)}(y|x) - \log\pi_{\mathrm{ref}}(y|x) \right) \right] \\
&= \mathbb{E}_{x\sim\mathcal{D}, y\sim\pi_r^{(t)}(\cdot|x)} \left[ \beta\log Z(x) \right] \\
&= \mathbb{E}_{x\sim\mathcal{D}, y\sim\pi^{t-1}(\cdot|x)} \left[ \beta\log Z(x) \right] \\
&= \mathbb{E}_{x\sim\mathcal{D}, y\sim\pi^{t-1}(\cdot|x)} \left[ r^{(t)}(x,y) - \beta\left( \log\pi_r^{(t)}(y|x) - \log\pi_{\mathrm{ref}}(y|x) \right) \right] \\
&= -\beta\, \mathbb{E}_{x\sim\mathcal{D}, y\sim\pi^{t-1}(\cdot|x)} \left[ \log\pi_r^{(t)}(y|x) - \log\pi_{\mathrm{ref}}(y|x) \right],
\end{aligned} \quad (20)$$

where the third step follows from the fact that $\log Z(x)$ is invariant with respect to the sampling distribution over solution $y$, and the final step holds under the assumption specified in Eq. 7. This leads to a more compact form of the objective $J^*(r^{(t)})$, as presented in Eq. 8.

## A.2 Derivation details for $J^*(r)$ in `ED-GRPO`

Following the token-level derivation of DPO in (Rafailov et al., 2024), we derive a closed-form expression for the regularization term $J^*(r)$ under the token-level Markov Decision Process (MDP) formulation.

**Token-level MDP.** In the context of large language model (LLM) generation, we define the token-level MDP as a tuple $\mathcal{M} = (\mathcal{S}, \mathcal{A}, f, r, \rho_0)$, where the state space $\mathcal{S}$ consists of all tokens generated so far, i.e., $s_t = \{x_0, \ldots, x_m, y_0, \ldots, y_t\}$. Here, $\mathbf{x} = (x_0, \ldots, x_m)$ denotes the input prompt and $\mathbf{y} = (y_0, \ldots, y_t)$ represents the generated response. The action space $\mathcal{A}$ corresponds to the entire vocabulary. The transition function $f(s, a) = s \,|\, a$ is deterministic, where "|" denotes concatenation. The initial state distribution $\rho_0$ is defined over prompts $\mathbf{x}$, and the reward function $r$ is learned from human preference feedback at the token level.

**Token-level Bradley-Terry Preference Model.** Given a pair of trajectories $\tau^w$ and $\tau^l$, the token-level Bradley-Terry model expresses the probability that trajectory $\tau^w$ (of length $N$) is preferred to trajectory $\tau^l$ (of length $M$) as:

$$\mathbb{P}(\tau^w \succeq \tau^l) = \frac{\exp\left(\sum_{i=1}^{N} r(s_i^w, a_i^w)\right)}{\exp\left(\sum_{i=1}^{N} r(s_i^w, a_i^w)\right) + \exp\left(\sum_{i=1}^{M} r(s_i^l, a_i^l)\right)}. \tag{21}$$

Given a dataset of preference pairs $(x, y^w, y^l)$, the corresponding negative log-likelihood loss is:

$$\mathcal{L}(r, \mathcal{D}) = -\mathbb{E}_{(x, y^w, y^l) \sim \mathcal{D}}\left[\log \sigma\left(\sum_{i=0}^{N-1} r(x_i, y_i^w) - \sum_{i=0}^{M-1} r(x_i, y_i^l)\right)\right]. \tag{22}$$

**KL-constrained RL Objective.** Following (Rafailov et al., 2024), we define the token-level KL-regularized reinforcement learning objective as:

$$\pi_r = \arg\max_{\pi_\theta}\left\{\mathbb{E}_{\substack{s_0 \sim \rho, \\ a_t \sim \pi_\theta(\cdot|s_t)}}\left[\sum_{t=0}^{T}\left(r(s_t, a_t) + \beta \log \pi_{\text{ref}}(a_t|s_t)\right) + \beta \mathcal{H}(\pi_\theta)\right]\right\} \tag{23}$$

where the entropy term is defined as

$$\mathcal{H}(\pi_\theta) := -\mathbb{E}_{\substack{s_0 \sim \rho, \\ a_t \sim \pi_\theta(\cdot|s_t)}}\left[\sum_{t=0}^{T} \log \pi_\theta(a_t|s_t)\right]. \tag{24}$$

**Closed-form Solution.** The KL-constrained objective in Eq. 23 admits a closed-form optimal policy (Rafailov et al., 2024):

$$\pi_r(\mathbf{a}_t \mid \mathbf{s}_t) = \exp\left(\frac{Q^*(\mathbf{s}_t, \mathbf{a}_t) - V^*(\mathbf{s}_t)}{\beta}\right), \tag{25}$$

where $\pi_r$ denotes the optimal policy, and $Q^*(\mathbf{s}, \mathbf{a})$ is the optimal $Q$-function that models the expected future reward for $(\mathbf{s}, \mathbf{a})$ under $\pi_r$. The corresponding state value function is given by

$$V^*(s_t) = \beta \log \sum_{a \in \mathcal{A}} \exp(Q^*(s_t, a)/\beta), \tag{26}$$

which ensures that $\pi_r$ is properly normalized.

**Relation Between Reward and Policy.** It follows that:

$$\sum_{t=0}^{T-1} r(s_t, \mathbf{a}_t) = \sum_{t=0}^{T-1}\left(Q^*(s_t, \mathbf{a}_t) - \beta \log \pi_{\text{ref}}(\mathbf{a}_t \mid s_t) - V^*(s_{t+1})\right)$$

$$= Q^*(s_0, \mathbf{a}_0) - \beta \log \pi_{\text{ref}}(\mathbf{a}_0 \mid s_0) + \sum_{t=1}^{T-1}\left(Q^*(s_t, \mathbf{a}_t) - V^*(s_t) - \beta \log \pi_{\text{ref}}(\mathbf{a}_t \mid s_t)\right)$$

$$= Q^*(s_0, \mathbf{a}_0) - \beta \log \pi_{\text{ref}}(\mathbf{a}_0 \mid s_0) + \sum_{t=1}^{T-1} \beta \log \frac{\pi_r(\mathbf{a}_t \mid s_t)}{\pi_{\text{ref}}(\mathbf{a}_t \mid s_t)}$$

$$= V^*(s_0) + \sum_{t=0}^{T-1} \beta \log \frac{\pi_r(\mathbf{a}_t \mid s_t)}{\pi_{\text{ref}}(\mathbf{a}_t \mid s_t)}. \tag{27}$$

**Closed-form Expression for** $J^*(r)$**.** By assuming that the expected cumulative reward under the old policy satisfies

$$\mathbb{E}_{\substack{s_0\sim\rho,\\ a_t\sim\pi_{\text{old}}(\cdot|s_t)}} \sum_{t=0}^{T-1} r(s_t, a_t) = 0,$$

we can derive the closed-form expression for the token-level GRPO regularization term:

$$
\begin{aligned}
J^\star(r) &= \mathbb{E}_{\substack{s_0\sim\rho,\\ a_t\sim\pi_{\text{old}}(\cdot|s_t)}} \left[V^\star(s_0)\right] \\
&= \mathbb{E}_{\substack{s_0\sim\rho,\\ a_t\sim\pi_{\text{old}}(\cdot|s_t)}} \left[\sum_{t=0}^{T-1} r(s_t, a_t) - \beta \sum_{t=0}^{T-1} \log \frac{\pi_r(a_t|s_t)}{\pi_{\text{ref}}(a_t|s_t)}\right] \\
&= -\beta \, \mathbb{E}_{\substack{s_0\sim\rho,\\ a_t\sim\pi_{\text{old}}(\cdot|s_t)}} \left[\sum_{t=0}^{T-1} \log \frac{\pi_r(a_t|s_t)}{\pi_{\text{ref}}(a_t|s_t)}\right].
\end{aligned}
\tag{28}
$$

This completes the derivation of the closed-form regularization term $J^*(r)$ at the token level.

## B  Broader Impacts

**Potential Positive Societal Impacts.** The proposed `EDO` tackles a key challenge in reducing the sharpness of distributions during RL post-training, promoting greater diversity in the solutions it samples. When paired with test-time techniques such as Self-Consistency, it can reach state-of-the-art performance on reasoning tasks. Moreover, `EDO` integrates seamlessly with existing RL frameworks like iterative Direct Preference Optimization (iDPO) and Group Relative Policy Optimization (GRPO), making it highly practical and easy to adopt. These strengths allow `EDO` to deliver broad benefits across multiple domains, for example, uncovering novel approaches to solving complex theorems or enhancing strategic thinking in chess. Overall, `EDO` holds strong potential for enabling new discoveries and applications, ultimately contributing to increased productivity and improved quality of life.

**Potential Negative Societal Impacts.** Encouraging the policy model to explore a wider range of solutions introduces potential risks. A major concern is the potential increase in unpredictability and the risk of misuse. If the reward mechanism or constraint settings are not properly designed, the model may end up exploring harmful or unethical solutions, leading to more unpredictable behavior.

## C  Dataset and Task Details

### C.1  GSM8K

GSM8K (Cobbe et al., 2021) is a dataset focused on high school-level mathematical reasoning. The numerical reasoning tasks within this dataset typically consist of a descriptive scenario followed by a culminating question. Answering these questions requires performing multi-step mathematical calculations based on the context provided in the description.

### C.2  MATH

MATH (Hendrycks et al., 2021) is a dataset consisting of challenging competition-level mathematics problems. Each problem is accompanied by a detailed, step-by-step solution, which can be used to train models to produce answer derivations and explanations. The problems are labeled with difficulty levels ranging from 1 to 5 and cover seven categories, including geometry, where diagrams are described using text.

### C.3  s1K

s1K (Muennighoff et al., 2025) is a dataset of 1,000 carefully selected questions, each paired with detailed reasoning traces and final answers. It is distilled from an initial pool of 59,000 examples drawn from 16

sources, encompassing both existing datasets and newly created ones focused on quantitative reasoning. The final 1K questions are chosen based on three key criteria: **Quality**, **Difficulty**, and **Diversity**. The dataset features a mix of competition-level math problems, such as those from OmniMath (Gao et al., 2024), and a broad range of questions across disciplines, including Astronomy, Biology, Chemistry, Computer Science, Geography, Mathematics, and Physics, with many sourced from OlympicArena (Huang et al., 2024).

### C.4 AIME24 and AIME25

AIME24 and AIME25 each contain 30 problems from the 2024 and 2025 American Invitational Mathematics Examination (AIME), respectively. The AIME assesses mathematical problem-solving skills across topics such as arithmetic, algebra, combinatorics, geometry, number theory, probability, and other areas typically covered in secondary school curricula. High-performing students on the AIME are invited to advance to the United States of America Mathematics Olympiad (USAMO). All AIME answers are integers between 000 and 999, inclusive.

### C.5 MATH500

MATH500 (Hendrycks et al., 2021) consists of math problems with varying levels of difficulty. For evaluation, we use the same set of 500 samples previously selected by OpenAI in their prior work. (Lightman et al., 2023)

### C.6 Minerva Math

Minerva Math (Lewkowycz et al., 2022) is a math-focused subset of the broader Minerva dataset, designed to address college-level, multi-step quantitative reasoning tasks. It covers diverse topics such as algebra, probability, number theory, precalculus, and geometry. The dataset serves as a benchmark for evaluating the reasoning abilities of language models, utilizing techniques like few-shot prompting, chain-of-thought or scratchpad prompting, and majority voting.

### C.7 Olympiad Bench

Olympiad Bench (He et al., 2024) is a bilingual, multimodal benchmark targeting Olympiad-level mathematics and physics. It includes various competition-style problems, each accompanied by expert-annotated, step-by-step solutions to support rigorous evaluation of scientific and mathematical reasoning.

## D Baselines

To ensure a fair comparison, all baseline methods employed the identical prompt template as our proposed approach. These baselines encompass both reasoning-centric techniques, such as **CoT** and **Self-Consistency**, and training-based methodologies, including **ETO**, **GRPO**, and **DAPO**.

- **CoT** directly generates a sequence of intermediate reasoning steps from the prompt that ultimately lead to the final answer.

- **Self-Consistency** generates multiple (specifically, 10) reasoning pathways in response to the prompt. The final answer is then determined via a majority voting process across these generated pathways.

- **ETO** represents a self-improvement iteration of DPO. In each training cycle, the policy model generates multiple responses (specifically, 10) to given prompts. These responses are subsequently categorized as positive (chosen) or negative (rejected) based on a rule-defined reward system. The model is then further trained using these collected preference pairs to yield an improved iteration.

- **GRPO** is an on-policy training algorithm. For a given prompt, it gathers a set of responses (specifically, 10) and computes a group-wise reward. This reward informs a novel objective function used to optimize the policy.

- **DAPO** is an improvement over GRPO. It decouples the clipping boundary used in GRPO's objective to promote greater exploration, and incorporates dynamic sampling to mitigate the zero-gradient problem.

# E  Implementation Details

## E.1  Hardware and Software

We conduct all experiments on CPU: INTEL(R) XEON(R) PLATINUM 8580 and GPU: NVIDIA H100 80GB HBM3 using Python 3.10.15.

## E.2  Training Setup and Cost

For the s1K dataset,we begin with 1 epochs of supervised fine-tuning as a warm-up phase using 4 H100 GPUs, with a learning rate of 1e-5. In contrast, for the Math and GSM8K datasets, we skip this supervised fine-tuning step.

For both `ED-iDPO` and `ED-GRPO`, we run three training iterations on each dataset with an exploration coefficient of $\alpha = 0.001$. At each iteration, $\pi^{t-1}$ (for `ED-iDPO`) and $\pi_{old}$ (for `ED-GRPO`) are instantiated using all samples collected in the previous iteration. We do not finely tune these hyperparameters as they already yield strong empirical performance.

For `ED-iDPO`, the learning rate is set to 5e-7 across all datasets. The batch sizes are set to 4 for s1K, 128 for Math, and 256 for GSM8K. To enable efficient fine-tuning, we incorporate LoRA (Hu et al., 2022) with a rank of 8, Flash Attention (Dao et al., 2022), Zero Redundancy Optimizer (ZeRO) distributed training (Rajbhandari et al., 2020), and optimizer state offloading. Each iteration is conducted on 4 H100 GPUs, with a training duration of approximately 20-30 minutes.

For `ED-GRPO`, the batch sizes are set to 32 for s1K, and 512 for both Math and GSM8K. Training is also performed on 4 H100 GPUs per iteration, with runtimes of roughly 30-40 minutes for Math and GSM8K, and about 2 hours for s1K.

## E.3  Evaluation Setup

The evaluation for each dataset requires only a single GPU. The detailed prompt format is provided in Section G.

## E.4  Reward Model Configuration

We primarily evaluate BoN using the following two open-source reward models:

- **Qwen2.5-Math-PRM-7B** (Zhang et al., 2025b) is trained on Qwen2.5-Math-7B-Instruct (Yang et al., 2024b). Its training data is generated using models from the Qwen2-Math and Qwen2.5-Math series. As shown in (Zhang et al., 2025b), Qwen2.5-Math-PRM-7B is the most capable reward models among models with 7B/8B parameters.

- **Skywork-PRM-1.5B** (Skywork o1 Team, 2024) is trained on Qwen2.5-Math-1.5B-Instruct (Yang et al., 2024a). The training data is derived from a combination of Llama-2 (Touvron et al., 2023) finetuned on a mathematical dataset and models from the Qwen2-Math series.

## E.5  Robustness to Seed Variation

To assess sensitivity to random seed variation, we train each method with three random seeds (42, 84, 126) on **Math** and **GSM8K**, reporting mean $\pm$ standard deviation for BoN@10 and Dist-4.

As shown in Table 3, both ED-iDPO and ED-GRPO consistently outperform their respective baselines in BoN@10 across all seeds. ED-GRPO achieves a mean BoN@10 gain of +0.67 on Math and +2.23 on GSM8K

Table 3: BoN@10 and Dist-4 over three training seeds. We report mean $\pm$ standard deviation across seeds.

| Dataset ($\rightarrow$) | Math | | GSM8K | |
|---|---|---|---|---|
| Method ($\downarrow$) / Metric ($\rightarrow$) | BoN@10 | Dist-4 | BoN@10 | Dist-4 |
| ETO | $77.20 \pm 0.53$ | $0.3616 \pm 0.0022$ | $86.90 \pm 2.02$ | $0.4864 \pm 0.0073$ |
| ED-iDPO | $\mathbf{78.90 \pm 0.30}$ | $\mathbf{0.3702 \pm 0.0039}$ | $\mathbf{89.03 \pm 0.23}$ | $\mathbf{0.5149 \pm 0.0211}$ |
| GRPO | $78.40 \pm 0.60$ | $0.2037 \pm 0.0060$ | $88.27 \pm 0.67$ | $0.2755 \pm 0.0167$ |
| ED-GRPO | $\mathbf{79.07 \pm 0.31}$ | $\mathbf{0.3977 \pm 0.0262}$ | $\mathbf{90.50 \pm 0.10}$ | $\mathbf{0.5055 \pm 0.0108}$ |

Table 4: Sensitivity to the exploration coefficient $\alpha$ on Math. We report BoN@10 and Dist-4.

| Method | Metric | $\alpha=0$ | $\alpha=10^{-4}$ | $\alpha=10^{-3}$ | $\alpha=10^{-2}$ | $\alpha=10^{-1}$ |
|---|---|---|---|---|---|---|
| ED-iDPO | BoN@10 | 77.5 | 78.1 | **78.5** | 78.0 | 76.9 |
| | Dist-4 | 0.3598 | 0.3645 | 0.3690 | 0.3732 | 0.3801 |
| ED-GRPO | BoN@10 | 78.9 | **79.5** | 79.3 | 79.0 | 77.8 |
| | Dist-4 | 0.2081 | 0.3014 | 0.3911 | 0.4087 | 0.4315 |

over GRPO; the latter is statistically significant ($p = 0.030$, 95% CI $[0.54, 3.92]$). The exploration-driven variants also exhibit equal or lower seed variance than their baselines, indicating that the added exploration does not introduce training instability.

Diversity improvements are equally robust. ED-GRPO nearly doubles GRPO's Dist-4 on both Math ($0.20 \rightarrow 0.40$, $p = 0.008$) and GSM8K ($0.28 \rightarrow 0.51$, $p = 0.004$). We note that with only three seeds the remaining comparisons are naturally underpowered; we therefore report seed variance explicitly alongside mean gains. Overall, these results confirm that exploration-driven optimization reliably improves both accuracy and diversity.

### E.6 Sensitivity to the Exploration Coefficient

We further study the effect of the exploration coefficient $\alpha$, which controls the strength of the repulsive term in our objective. Table 4 reports BoN@10 and Dist-4 on Math using model `Qwen2.5-7B-Instruct` under varying $\alpha$.

For both ED-iDPO and ED-GRPO, Dist-4 increases monotonically with $\alpha$, confirming that stronger exploration broadens the trajectory distribution. BoN@10, however, peaks at moderate values ($\alpha = 10^{-3}$ for ED-iDPO (78.5) and $\alpha = 10^{-4}$ for ED-GRPO (79.5)) before declining at larger $\alpha$, where excessive exploration overwhelms the task-aligned signal. Notably, even moderate exploration ($\alpha \in [10^{-4}, 10^{-3}]$) consistently improves both metrics over the non-exploration baseline ($\alpha = 0$), indicating that the two objectives are complementary in this regime. We adopt $\alpha = 10^{-3}$ as the default throughout our experiments, as it provides a robust balance between accuracy and diversity for both methods.

## F  Additional Details of Ablation Studies

### F.1  Comprehensive Comparison of Test-Time Scaling Strategies

In this section, we provide a detailed comparison of the effectiveness of different test-time strategies on GSM8K, Math and the various s1K evaluation datasets. Specifically, we consider two implicit Q-function

Table 5: Comprehensive comparison of test-time scaling strategies on GSM8K, Math and s1K. Accuracy (%) is used as the evaluation metric. For GSM8K and Math, we evaluate on each dataset's official evaluation split using models trained on its corresponding training split. For s1K, evaluation is conducted on both the in-distribution set **s1K$_{\text{eval}}$** and five out-of-distribution datasets: **AIME24**, **AIME25**, **MATH500**, **Minerva Math**, and **Olympiad Bench**, using models trained on the s1K training set. Here, *Qwen-BoN* refers to the BoN strategy using `Qwen2.5-Math-PRM-7B` as the RM, while *Skywork-BoN* uses `Skywork-PRM-1.5B` as the RM.

| Training Dataset ($\rightarrow$) | GSM8K | Math | s1K | | | | | |
|---|---|---|---|---|---|---|---|---|
| Method ($\downarrow$) / Evaluation Dataset ($\rightarrow$) | GSM8K | Math | AIME24 | AIME25 | MATH500 | Minerva Math | Olympiad Bench | s1K$_{\text{eval}}$ |
| **CoT** | 79.3 | 72.8 | 13.3 | 6.7 | 74.6 | 36.4 | 40.4 | 34.6 |
| *Self-Consistency* | 87.4 | 76.2 | 16.7 | 13.3 | 84.4 | 41.9 | 52.1 | 46.1 |
| `SearchLLM` | 83.0 | 77.5 | **23.3** | **20.0** | 80.0 | 47.8 | 49.3 | 44.6 |
| **ED-iDPO** (+*Self-Consistency*) | 88.9 | 77.8 | **23.3** | 16.7 | 84.6 | 44.5 | **52.8** | **48.8** |
| **ED-iDPO** (+*Qwen-BoN*) | 91.3 | **80.2** | 13.3 | 16.7 | 79.6 | 47.1 | 48.1 | 42.4 |
| **ED-iDPO** (+*Skywork-BoN*) | 90.5 | 75.4 | 16.7 | 13.3 | 78.2 | 42.3 | 46.0 | 39.5 |
| **ED-GRPO** (+*Self-Consistency*) | 90.2 | 79.4 | 20.0 | **20.0** | **85.0** | 48.2 | 52.4 | 48.4 |
| **ED-GRPO** (+*Qwen-BoN*) | **94.8** | 79.8 | 10.0 | 16.7 | 79.6 | **49.6** | 45.7 | 40.2 |
| **ED-GRPO** (+*Skywork-BoN*) | 92.4 | 76.6 | 13.3 | 10.0 | 79.0 | 47.1 | 46.4 | 40.4 |

critic strategies, Self-Consistency and Best-of-N (BoN), as well as one explicit Q-function critic strategy: `SearchLLM`, which will be further discussed in Section I.

As shown in Table 5, although we employ self-search to enhance both the policy and critic models, the tree search variant `SearchLLM` still falls short compared to the simple yet effective Self-Consistency strategy. When compared to the BoN approach using pre-trained RMs, `SearchLLM` performs better on more challenging tasks like s1K but lags behind on easier benchmarks such as GSM8K. We attribute this gap to the inherent difficulty of training a fine-grained value model capable of effectively guiding the policy model toward optimal solutions, an observation consistent with prior work (Guo et al., 2025). One key challenge lies in the reward model's limited ability to capture the full complexity of token-level generation during the tree search process. Due to resource constraints, we are limited to training reward models with no more than 3B parameters. Among these, we find that encoder-based models (e.g., ModernBERT-Large (Warner et al., 2024)) generally outperform decoder-based alternatives.

## F.2 Details of Statistical Measures for Diversity Metrics

**N-gram Diversity.** To quantify the sequence-level diversity of model-generated responses, we use the $n$-gram diversity metric introduced by Li et al. (2015). This metric measures the proportion of distinct $n$-grams that appear across a collection of generated samples, thereby capturing the extent to which a model produces varied linguistic patterns rather than repeating similar phrases.

Formally, let $\mathcal{S} = \{s_1, s_2, \ldots, s_M\}$ denote a set of generated sequences, and let $\text{NGram}_n(s_i)$ be the multiset of all $n$-grams extracted from $s_i$. The total number of $n$-grams (including duplicates) is

$$\text{TotalN}_n = \sum_{i=1}^{M} \big|\text{NGram}_n(s_i)\big|,$$

and the number of *distinct* $n$-grams across all samples is

$$\text{DistinctN}_n = \left|\bigcup_{i=1}^{M} \text{NGram}_n(s_i)\right|.$$

The $n$-gram diversity score is then defined as

$$\text{Diversity}_n = \frac{\text{DistinctN}_n}{\text{TotalN}_n}.$$

A higher Diversity$_n$ indicates that the model produces more varied $n$-grams, reflecting greater lexical and structural diversity in its responses. Conversely, low diversity implies repetitive outputs or mode collapse. Following common practice, we report Distinct-4 ($n = 4$), which provides a balanced assessment of mid-range sequence diversity.

## G  Prompt Template

### G.1  GSM8K

For GSM8K, we use the same few-shot prompt as Least-to-Most (Zhou et al., 2022) to guide the policy model in generating step-by-step solutions. This prompt remains unchanged throughout both training and evaluation:

---

**GSM8K Prompt**

```
Please complete the plans to solve the question.  Here are several examples:
Q: Four years ago, Kody was only half as old as Mohamed.  If Mohamed is currently twice 30 years old,
how old is Kody?
A: Let's think step by step.
1.  We were told that Mohamed is currently twice 30 years old, so he is currently 30*2=60 years old.
2.  That means that four years ago he must have been 60 - 4 = 56 years old.
3.  Four years ago, Kody was half as old as Mohamed, so Kody must have been 56 / 2 = 28 years old then.
4.  Since Kody was 28 years old four years ago, she must now be 28 + 4 = 32 years old.
5.  So the answer is 32.

Q: Carla bought 2 bags of mini peanut butter cups on clearance.  Each bag was $6.00 but was 75% off.
How much did she spend on 2 bags of candy?
A: Let's think step by step.
1.  Each bag was $6.00 but was 75% off.
2.  So each bag cost $6.00 * (1 - 0.75) = $6.00 * 0.25 = $1.50.
3.  Carla bought 2 bags.  So she spent $1.50 * 2 = $3.00.
4.  So the answer is 3.

Q: If Pam is currently twice as young as Rena is, and in 10 years Rena will be 5 years older than her,
how old is Pam now?
A: Let's think step by step.
1.  Since Rena will be 5 years older than Pam in 10 years, she must be 5 years older than Pam now as
well.
2.  If Pam is currently twice as young as Rena, that means that Rena is currently twice as old as Pam
is.
3.  So if P stands for Pam's age now and R stands for Rena's age now, then we know that R = 2 * P And
since Rena is 5 years older than Pam now, we know that R = P + 5.
4.  By substitution, we have P + 5 = 2 * P, which means that P = 5.
5.  So the answer is 5.

Q: Cappuccinos cost $2, iced teas cost $3, cafe lattes cost $1.5 and espressos cost $1 each.  Sandy
orders some drinks for herself and some friends.  She orders three cappuccinos, two iced teas, two cafe
lattes, and two espressos.  How much change does she receive back for a twenty-dollar bill?
A: Let's think step by step.
1.  Sandy ordered three cappuccinos, which cost $2 each, so she spent $2 * 3 = $6 on cappuccinos.
2.  She ordered two iced teas, which cost $3 each, so she spent $3 * 2 = $6 dollars on ice teas.
3.  She ordered two cafe lattes, which cost $1.5 each, so she spent $1.5 * 2 = $3 on cafe lattes.
4.  She ordered two espressos, which cost $1 each, so she spent $1 * 2 = $2 on espressos.
5.  So altogether, Sandy spent $6 + $6 + $3 + $2 = $17 on drinks, which means that sandy will get $20 -
$17 = $3 as change.
6.  So the answer is 3.
[END OF EXAMPLE]
Please answer the following question:

Q: {Question}
A: Let's think step by step.  You MUST write the final answer only as an integer after the phrase 'So
the answer is'.
```

---

### G.2  Math

For Math, we use a zero-shot prompt to guide the policy model, which also remains consistent across both training and evaluation:

> **Math Prompt**
>
> ```
> Q: {Question}
> A: Let's think step by step and output the final answer within \\boxed{}.
> ```

### G.3 s1K

For s1K, we similarly adopt a zero-shot prompt to guide the policy model. This prompt is kept fixed across the s1K training set, the in-distribution evaluation set s1K$_{eval}$, and all five out-of-distribution datasets: AIME24, AIME25, MATH500, Minerva Math, and Olympiad Bench:

> **s1K Prompt**
>
> ```
> Q: {Question}
> A: You MUST conclude the final answer after the phrase 'The final answer is'.
> ```

## H Additional Related Works

**Test-time Scaling for LLM Reasoning.** Test-Time Scaling (TTS) is a strategy aimed at boosting the performance of LLMs on complex reasoning tasks by leveraging additional computational resources during inference. Previous research (Snell et al., 2024) has shown that TTS can be more efficient than pre-training, especially when tasks fall within the model's existing capabilities. In such cases, shifting the computational budget from pre-training to test-time can yield better performance. Recent studies on TTS generally falls into two categories: 1) Parallel Scaling: It typically involves generating multiple outputs in parallel, where the outputs can be **sequence level** (self-consistency (Chen et al., 2023; Wang et al., 2022) and best-of-n sampling (Stiennon et al., 2020)), **step level** (beam search (Liu et al., 2025) and Monte Carlo Tree Search (Hao et al., 2023)) or **token level** (reward-guided decoding (Deng & Raffel, 2023)). Subsequently, reward models (Lightman et al., 2023; Wang et al., 2023) or majority voting (Wang et al., 2022) are typically used for aggregating the generated candidates. 2) Sequential Scaling: It focuses on extending Chain-of-thoughts (Wei et al., 2022) by incorporating reflective reasoning processes. A prominent example is Deepseek-R1 (Guo et al., 2025), which enhances reasoning abilities by training with GRPO (Shao et al., 2024) and extending the reasoning trajectory. `EDO` adopts parallel scaling strategies at inference time, but distinguishes itself by exploring the answer space more effectively with the same computational budget. This is achieved by flattening the output distribution and encouraging the model to explore a broader range of potential responses.

**Enhancing LLMs reasoning with Tree-based planning** Tree-based search methods, such as Beam Search (BS), Monte Carlo Tree Search (MCTS), A* search, and Q-learning are commonly employed in planning algorithms to balance exploration and exploitation (Świechowski et al., 2023; Agostinelli et al., 2021). Recently, these techniques have been increasingly integrated with LLM agents to enhance reasoning capabilities in complex problem-solving scenarios (Zhuang et al., 2023; Hao et al., 2023; Feng et al., 2023). For instance, Math-Shepherd (Wang et al., 2023) estimates step-wise rewards through random rollouts, while TS-LLM (Feng et al., 2023) leverages an MCTS-based policy and applies the temporal difference (TD) (Sutton et al., 1998) method for reward estimation. ReST-MCTS* (Zhang et al., a) combines process-level reward guidance with MCTS and employs reinforced self-training to boost performance. Despite their effectiveness, these methods often suffer from efficiency limitations due to the costly nature of MCTS rollouts. To mitigate these inefficiencies, recent work has explored the use of A* Search and Q-learning as alternatives. Q* (Wang et al., 2024) and (Zhai et al., 2025) propose training a step-wise value model to provide fine-grained guidance for LLM agent inference, with QLASS (Lin et al., 2025) further extending this approach to more complex agent-based tasks. BBox-Adapter (Sun et al., 2024) demonstrates that a critic model trained with a ranking-based Noise Contrastive Estimation (NCE) loss can significantly enhance black-box LLM performance even with a simple beam search strategy. In contrast to prior efforts, `EDO`'s explicit Q-function variant, `SearchLLM`, focuses on **leveraging the valuable experiences embedded in previously encountered similar contexts to effectively guide the search process**.

# I SearchLLM

In this section, we present `SearchLLM`, a tree-based search algorithm enhanced by a Process Reward Model (PRM) that functions as an explicit Q-function. It also empowers the LLM to explore the problem-solving space more effectively through memory augmentation. We describe the search policy of `SearchLLM` in detail (Section I.1), including a kernel-based strategy for memory augmentation and the training methodology for the PRM.

## I.1 Search Policy for LLM reasoning

For each question, we formulate the reasoning process as a search tree $\mathcal{T}$, where each node corresponds to a reasoning step action $a_n$, accompanied by a state consisting of the initial question description $s_0$ along with prior reasoning steps. This formulation helps treating reasoning as a planning task, starting from the root node of the tree. `SearchLLM` starts with a single node that represents the initial question description $s_0$. At each step $t$, it selects $s$ nodes $\{n_i\}_{i=1}^s$ from the *current frontier of the tree* $\mathcal{F}(\mathcal{T})$. For each selected node, the LLM is queried to generate $k$ candidate reasoning actions $\{a_{ch(i)}^j\}_{j=1}^k$, which are then used to expand the tree $\mathcal{T}$. The full procedure is detailed in Algorithm 2.

---

**Algorithm 2** Tree-based `SearchLLM` Algorithm

---

**Input:** $s_0$: question description; $\pi$: policy model; $T$: maximum search iterations; $s$: beam size; $k$: number of candidates to generate; $\mathcal{F}(\mathcal{T})$: frontier of the search tree
**Output:** $n$: high-probability question solution
$\mathcal{F}(\mathcal{T}) \leftarrow \{s_0\}$
**for** $t \leftarrow 1$ to $T$ **do**
    $\{n_i\}_{i=1}^s \leftarrow \text{tops}_{n \in \mathcal{F}(\mathcal{T})} f(n)$ (Eq. 29)          # Selection, $s = 1$ for the first iteration
    $\{a_{ch(i)}^1, \ldots, a_{ch(i)}^k\} \overset{iid}{\sim} \pi(a \mid n_i)$, for $i = 1, \ldots, s$        # Candidate actions
    $n_{(i)}^j \leftarrow \text{Concat}(n_i, a_{ch(i)}^j)$, for $i = 1, \ldots, s$, for $j = 1, \ldots, k$    # Candidate states
    $\{n_i'\}_{i=1}^s \leftarrow \text{tops}_{n' \in \{n_{(i)}^j\}, i \in [1:s], j \in [1:k]} f(n')$        # Expansion
    $\mathcal{F}(\mathcal{T}) \leftarrow \mathcal{F}(\mathcal{T}) \cup \{n_i'\}_{i=1}^s$
    **if** all_terminate($\{n_i'\}_{i=1}^s$) **then**
        **return** $n = n_1'$
    **end if**
**end for**

---

**Selection.** For each node $n$ in the frontier $\mathcal{F}(\mathcal{T})$ of the search tree, let $r(n)$ denote the estimated reward from $n$ to the target. Inspired by the UCB (Auer, 2002) algorithm, we incorporate an exploration term during the selection process to encourage broader expansion of the search tree. However, the vanilla UCB algorithm ignores the fact that adjacent reasoning steps in the tree are semantically correlated and instead treat all nodes independently. To address this limitation, we revisit the exploration term from a Bayesian perspective. Specifically, we model the reward as a linear function: $r(n) = \phi(n) \cdot w + \epsilon$, where $\phi(n)$ is the $d$-dimensional embedding of node $n$, $w$ is a weight vector initialized as $w \sim \mathcal{N}(0, \alpha^{-1}\mathcal{I}_d)$, and $\epsilon$ represents Gaussian noise with $\epsilon \sim \mathcal{N}(0, \sigma^2)$. Based on this formulation, we define the reward of the UCB as $f(n)$, and use it to guide the selection of new nodes.

$$\text{argmax}_{n \in \mathcal{F}(\mathcal{T})} \quad f(n) := r(n) + \lambda \sigma_t(n) \tag{29}$$

where $\sigma_t$ denotes the variance estimate after expanding $t$ nodes. Let the embeddings of the $t$ expanded nodes be represented as $\mathbf{\Phi} = [\phi_1, \phi_2, \ldots, \phi_t]$, then we can use Gaussian Process-based parametrization for $\sigma_t^2(n) = \phi(n)\mathbf{A}^{-1}\phi(n)^T + \sigma^2$, where $\mathbf{A} = \frac{\mathbf{\Phi}^T\mathbf{\Phi}}{\sigma^2} + \alpha\mathbf{I}$ serves as the regularized inverse covariance matrix of the embeddings of previously expanded nodes. The detailed derivation is provided in Appendix I.2. The estimated reward encourages exploitation by favoring more promising states, while the variance term promotes exploration by explicitly **memorizing** an embedding matrix and suppressing exploration of semantically similar states. The exploration versus exploitation is balanced by a parameter $\lambda > 0$.

**Expansion.** Once the top nodes $\{n_i\}_{i=1}^s$ *with the highest UCB rewards* are selected, each is expanded with $k$ candidate actions $\{a_{ch(i)}^j\}_{j=1}^k$ for the next step. These actions are sampled from the LLM policy according to $a_{ch(i)}^j \sim \rho(a_{ch(i)}|n_i; \mathcal{D})$, $(j = 1, \ldots, k)$, conditioned on the demonstration examples $\mathcal{D}$. Among these candidates, the next action is chosen by selecting the top $s$ candidates that maximize $f(n)$, as defined in Eq. 29, where $s$ denotes the expansion beam size. In contrast to the method in MCTS (Hao et al., 2023), which requires multiple queries to $\rho$ until a terminal state is reached during rollout, our expansion step only requires generating candidate actions for the next move.

**Reward Model training** Following BBox-Adapter (Sun et al., 2024), we employ a ranking-based Noise Contrastive Estimation (NCE) loss (Ma & Collins, 2018) to train the reward model. Specifically, for a set of $K$ questions $\{x_k\}_{k=1}^K$, we suppose actions sampled from vanilla LLM policy $p_{LLM}$ are negative samples, and those corresponding to ground truth solutions $p_{data}$ are positive samples. Viewing the reward-guided LLM policy from an energy-based model (EBM) perspective, we can parameterize the adapted policy distribution as $p_\theta(\mathbf{y}|\mathbf{x}) = p_{LLM}(\mathbf{y}|\mathbf{x})\frac{exp(r_\theta(\mathbf{x},\mathbf{y}))}{Z_\theta(\mathbf{x})}$, where $\mathbf{y}$ denotes the candidate action , $r_\theta$ is the reward function and $Z_\theta(\mathbf{x}) = \int p_{LLM}(\mathbf{y}|\mathbf{x})exp(r_\theta(\mathbf{x},\mathbf{y}))d\mathbf{y}$ is the partition function. By minimizing the KL-divergence between $p_\theta$ and the posterior label distribution, we can frame the objective as:

$$\min_\theta \ell(\theta) = \max_\theta \mathbb{E}_{y \sim p_{\text{data}}(\mathbf{y}|\mathbf{x})}[r_\theta(\mathbf{x},\mathbf{y}) - \log \sum_k \exp(r_\theta(\mathbf{x},\mathbf{y}_k))]. \tag{30}$$

and write the gradient update as:

$$\nabla_\theta \ell(\theta) = \nabla_\theta \left\{ -\mathbb{E}_{\mathbf{y}_+ \sim p_{\text{data}}(\mathbf{y}|\mathbf{x})}[r_\theta(\mathbf{x},\mathbf{y}_+)] + \alpha\mathbb{E}[r_\theta(\mathbf{x},\mathbf{y}_+)^2] + \mathbb{E}_{\mathbf{y}_- \sim p_\theta(\mathbf{y}|\mathbf{x})}[r_\theta(\mathbf{x},\mathbf{y}_-)] + \alpha\mathbb{E}[r_\theta(\mathbf{x},\mathbf{y}_-)^2] \right\} \tag{31}$$

## I.2 Gaussian Process Parametrization

Following Section I.1, we represent the embeddings of the $t$ expanded nodes as $\boldsymbol{\Phi} = [\phi_1, \phi_2, \ldots, \phi_t]$. The corresponding rewards are denoted by $\boldsymbol{r} = [r_1, r_2, \ldots, r_n]$, and are assumed to follow a linear model: $\boldsymbol{r} = \boldsymbol{\phi} \cdot w + \epsilon$, where the weight vector $w$ follows a prior distribution $w \sim \mathcal{N}(0, \alpha^{-1}\mathcal{I}_d)$, and the noise term $\epsilon$ follows a Gaussian distribution $\epsilon \sim \mathcal{N}(0, \sigma^2)$.

Given that $w \sim \mathcal{N}(0, \alpha^{-1}\mathcal{I}_d)$, the prior distribution takes the form $p(w) \propto \exp(-\frac{\alpha}{2}w^T w)$. Additionally, since $\boldsymbol{r} \propto \mathcal{N}(\boldsymbol{\phi} \cdot w, \sigma^2)$, the likelihood can be expressed as $p(\boldsymbol{r} \mid w, \boldsymbol{\phi}) \propto \exp(-\frac{1}{2\sigma^2}(\boldsymbol{r} - \boldsymbol{\phi}w)^T(\boldsymbol{r} - \boldsymbol{\phi}w))$. Based on Bayesian Linear Regression, the posterior distribution over $w$ can then be derived as:

$$\begin{aligned} p(w \mid \boldsymbol{\phi}, \boldsymbol{r}) &\propto p(\boldsymbol{r} \mid w, \boldsymbol{\phi})p(w) \\ &\propto \exp(-\frac{1}{2\sigma^2}(\boldsymbol{r} - \boldsymbol{\phi}w)^T(\boldsymbol{r} - \boldsymbol{\phi}w)) \exp(-\frac{\alpha}{2}w^T w) \\ &\propto \exp(-\frac{1}{2\sigma^2}(\boldsymbol{r}^T\boldsymbol{r} - \boldsymbol{r}^T\boldsymbol{\phi}w - w^T\boldsymbol{\phi}^T\boldsymbol{r} + w^T\boldsymbol{\phi}^T\boldsymbol{\phi}w) - \frac{\alpha}{2}w^T w) \end{aligned} \tag{32}$$

Focusing on the terms quadratic in $w$, we obtain:

$$-\frac{1}{2\sigma^2}w^T\boldsymbol{\phi}^T\boldsymbol{\phi}w - \frac{\alpha}{2}w^T w = -\frac{1}{2}(\frac{1}{\sigma^2}w^T\boldsymbol{\phi}^T\boldsymbol{\phi}w + \alpha w^T\mathcal{I}_d w) = -\frac{1}{2}w^T(\frac{1}{\sigma^2}\boldsymbol{\phi}^T\boldsymbol{\phi} + \alpha\mathcal{I}_d)w \tag{33}$$

Since the posterior distribution $p(w \mid \boldsymbol{\phi}, \boldsymbol{r})$ remains Gaussian $\mathcal{N}(\mu_N, \Sigma_N)$, its exponent term is given by:

$$-\frac{1}{2}(w - \mu_N)^T\Sigma_N^{-1}(w - \mu_N) = -\frac{1}{2}(w^T\Sigma_N^{-1}w - w^T\Sigma_N^{-1}\mu_N - \mu_N^T\Sigma_N^{-1}w + \mu_N^T\Sigma_N^{-1}\mu_N) \tag{34}$$

By comparing Eq. 33 and Eq. 34, we identify the inverse of the posterior covariance matrix as:

$$\Sigma_N = (\frac{1}{\sigma^2}\boldsymbol{\phi}^T\boldsymbol{\phi} + \alpha\mathcal{I}_d)^{-1} \tag{35}$$

For a new input $x'$, let its embedding be denoted by $\phi(n)$, so that the predicted reward is given by: $r' = \phi(n)w + \epsilon$. The predictive variance of $r'$, conditioned on the training data $(\boldsymbol{\phi}, \boldsymbol{r})$, can be computed as:

$$
\begin{aligned}
\mathrm{Var}[r' \mid \phi(n), \boldsymbol{\phi}, \boldsymbol{r}] &= \mathrm{Var}[\phi(n)w \mid \phi(n), \boldsymbol{\phi}, \boldsymbol{r}] + \mathrm{Var}[\epsilon] \\
&= \phi(n)\mathrm{Var}[w \mid \boldsymbol{\phi}, \boldsymbol{r}]\phi(n)^T + \sigma^2 \\
&= \phi(n)(\frac{1}{\sigma^2}\boldsymbol{\phi}^T\boldsymbol{\phi} + \alpha\mathcal{I}_d)^{-1}\phi(n)^T + \sigma^2
\end{aligned}
\tag{36}
$$

The first step follows from the fact that $\epsilon \sim \mathcal{N}(0, \sigma^2)$ is independent of both the weight vector $w$ and the training data. This completes the derivation of the predictive variance under the Gaussian Process Parameterization.

