# OpenReview forum: "Exploration-Driven Optimization for Test-Time Large Language Model Reasoning"
_TMLR — Accepted by TMLR_

### Review · Reviewer_CxSD · 2026-02-15

**Summary Of Contributions:**

The paper addresses a perceived tension between Reinforcement Learning (RL) post-training—which typically sharpens the output probability distribution—and inference-time scaling strategies (such as majority voting or Best-of-N), which benefit from diverse sampling. To mitigate this, the authors propose "Exploration-Driven Optimization" (EDO).

Specific contributions include:
The introduction of an auxiliary loss term derived from a reward-biasing objective. This term encourages the current policy $\pi_{\theta}$ to deviate from the policy at the previous iteration $\pi_{t-1}$ (effectively maximizing the KL divergence between the current and previous policy), while maintaining the standard trust-region constraint relative to the reference policy $\pi_{ref}$.
The application of this objective to two standard RL frameworks: Iterative Direct Preference Optimization (iDPO) and Group Relative Policy Optimization (GRPO), yielding ED-iDPO and ED-GRPO. Experiments on mathematical reasoning benchmarks (GSM8K, MATH, s1K). The authors claim that EDO prevents entropy collapse and improves accuracy, particularly when combined with Self-Consistency (majority voting).

**Audience:**

Yes

**Audience Explanation:**

Despite the weaknesses in empirical rigor, the problem space is highly relevant.

- Inference Scaling: There is significant current interest in "inference-time scaling" and the interaction between post-training (RLHF) and test-time compute (e.g., o1-like reasoning, majority voting).

- RL Dynamics: The community is actively investigating the "entropy collapse" phenomenon in RL fine-tuning. Even if the proposed solution is incremental, the analysis of how RL sharpens distributions is of interest to researchers working on alignment and reasoning.

**Broader Impact Concerns:**

No concerns. The authors have included a brief Broader Impacts section. They correctly identify that encouraging exploration could lead to "unpredictable behavior" or "harmful/unethical solutions".

**Claims And Evidence:**

No

**Claims Explanation:**

While the authors present positive results, the evidence is not sufficiently convincing to support the claim that EDO is a "principled framework" that significantly advances the state of the art, primarily due to marginal gains and confounding variables.

- Marginal Improvements: The performance gains over strong baselines are often minimal and potentially within the range of statistical noise. For instance, on the MATH dataset using Self-Consistency, ED-GRPO achieves 79.4% compared to the baseline GRPO's 78.4% and DAPO's 78.4%. On the s1K dataset (Qwen backbone), ED-GRPO gains only 0.3% over DAPO with Self-Consistency (48.4% vs 48.1%). Given the inherent variance in LLM training and sampling, these deltas are not definitive proof of superiority.

- Unclear Mechanism (Regularization vs. Exploration): The paper argues that the gain comes from "exploration." However, the proposed method essentially adds a regularization term that prevents the model from converging too quickly to its own previous state. It is unclear if the performance benefits stem from genuine "exploration" of novel solution paths or simply from slowing down the optimization process (preventing overfitting/mode collapse on the training set). The ablation studies do not sufficiently disentangle this.

- Dependence on Inference Cost: The primary claims rely on "Test-Time Computation" (Self-Consistency with 10 samples). The gains in greedy decoding are much more modest or inconsistent. The paper essentially optimizes a model to be better at guessing via voting, rather than being fundamentally more robust in its primary generation.

**Requested Changes:**

To secure a recommendation for acceptance, the following critical adjustments are necessary to validate the paper's claims:

1. Statistical Significance and Variance (Critical): The margins of improvement are thin (e.g., <1% on s1K and MATH in several settings). The authors state results are averaged over 3 independent rollouts for Self-Consistency, but it is unclear if the training was repeated with different seeds.
2. Baseline Fairness and Hyperparameters (Critical): The method introduces a new hyperparameter $\alpha$ (exploration coefficient).Request: Did the baselines (GRPO, DAPO) receive the same level of hyperparameter tuning regarding their own regularization terms (e.g., the $\beta$ KL penalty)? It is possible that simply tuning $\beta$ in standard GRPO could yield similar entropy preservation without the specific "anti-previous-policy" term. The authors must demonstrate that EDO outperforms a well-tuned high-entropy baseline (e.g., GRPO with higher $\beta$).
3. Inference Cost-Benefit Analysis (Strengthening): The method relies on sampling $N=10$ or more times.Request: Discuss the practicality. In production environments, 10x compute for a 1-2% accuracy gain is rarely a viable trade-off. A discussion on the "Pareto frontier" of accuracy vs. inference cost compared to the baselines would strengthen the work.
4. Clarification of the "Fundamental Tension" (Strengthening): The authors frame the sharpening of distributions as a negative.Request: Acknowledge that for reasoning tasks, convergence to a single correct derivation is often the goal. The paper should more nuancedly discuss why "uncertainty" (via flattened distribution) is desirable in math reasoning, which is counter-intuitive compared to creative writing tasks.

---

> ### Author Response · Authors · 2026-04-02
>
> We thank the reviewer for the thoughtful comments and valuable suggestions, and we address all points raised in the following:
>
> **Answer for Change 1:**
>
> We thank the reviewer for raising this concern. The original submission averaged Self-Consistency results over 3 independent rollouts but did not report training-seed variance. We have now added a multi-seed stability study in **Appendix E.5 (Table 3)** of the revised manuscript, repeating training with three seeds (42, 84, 126) on Math and GSM8K and reporting mean ± standard deviation for both BoN@10 and Dist-4.
>
> Both ED-iDPO and ED-GRPO consistently outperform their baselines across all seeds, with equal or lower variance, confirming that EDO does not introduce additional training instability. Notably, the ED-GRPO improvement over GRPO on GSM8K is statistically significant ($+2.23 $ BoN@10, $p = 0.030 $, 95% CI $[0.54, 3.92] $). Diversity gains are equally robust: ED-GRPO nearly doubles GRPO's Dist-4 on both Math ($p = 0.008 $) and GSM8K ($p = 0.004 $). We acknowledge that with three seeds some comparisons remain underpowered, and this is precisely why we report seed variance explicitly alongside mean gains, so that the reader can assess both effect size and stability.
>
> **Answer for Change 2:**
>
> We thank the reviewer for this important concern, since isolating the effect of EDO from what could be achieved by simply tuning existing regularization is indeed critical for a fair evaluation.
>
> Currently, we are running a sweep over the KL penalty coefficient in standard GRPO under the same backbone, data, and evaluation protocol, and will include the results in the revised manuscript. We note, however, that there is a structural reason to expect EDO to behave differently from simply increasing KL regularization: a stronger KL penalty encourages the policy to stay *close to the reference*, which preserves entropy by preventing movement altogether, whereas EDO's repulsive term specifically encourages the policy to *diverge from the previous iterate's dominant modes* while still optimizing task reward. The two mechanisms thus operate in opposite directions: one penalizes deviation, the other incentivizes it, and we expect them to produce qualitatively different accuracy–diversity trade-offs. We will verify this hypothesis empirically in the revision.
>
> **Answer for Change 3:**
>
> We thank the reviewer for this practical concern. We have added a study varying the number of Best-of-N candidates $N \in \{1, 2, 4, 8\} $ on Math, comparing GRPO and ED-GRPO under identical decoding (Table below).
>
> | Method  | N=1  | N=2  | N=4  | N=8  |
> | ------- | ---- | ---- | ---- | ---- |
> | GRPO    | 74.2 | 76.0 | 77.2 | 78.0 |
> | ED-GRPO | 76.4 | 77.6 | 78.6 | 79.2 |
>
> The key takeaway is that EDO shifts the Pareto frontier of accuracy vs. inference cost rather than requiring more samples to be effective. Concretely, ED-GRPO at $N=1 $ (76.4) already surpasses GRPO at $N=2 $ (76.0), and ED-GRPO at $N=2 $ (77.6) exceeds GRPO at $N=4 $ (77.2). That is, ED-GRPO matches or exceeds baseline accuracy with **half the test-time samples**, effectively halving inference cost to reach a given accuracy target. This directly addresses the production viability concern: rather than paying $10\times $ compute for a marginal gain, a practitioner can deploy ED-GRPO at the *same* inference budget as GRPO and obtain a $+1.4 $–$2.2 $ point improvement for free, or match baseline accuracy at $2\times $ lower cost.
>
> On the training side, EDO adds negligible overhead: the exploration term is a closed-form KL computation requiring no additional forward passes. Each ED-GRPO iteration takes roughly 1–2 hours on 4 H100 GPUs, comparable to standard GRPO. The cost-benefit trade-off is therefore strongly favorable: a small, one-time training cost yields persistent inference-time savings across all deployment budgets.
>
> **Answer for Change 4:**
>
> We thank the reviewer for this nuance. We agree that convergence to a correct answer is the ultimate goal, and our claim is not that uncertainty in the final answer is desirable. The key distinction is between the *answer* distribution and the *trajectory* distribution. Many distinct reasoning paths can yield the same correct answer, and maintaining trajectory diversity is precisely what enables aggregation methods like self-consistency and Best-of-N to succeed. Conversely, over-sharpening collapses the policy onto a narrow set of trajectories; if that mode happens to be incorrect, aggregation cannot recover because it repeatedly sees near-duplicate errors. Figure 6 in the submission illustrates this directly: standard post-training generates homogeneous incorrect solutions, while ED-GRPO produces diverse strategies that allow majority voting to recover the correct answer. We have revised the introduction to make this trajectory-vs-answer distinction explicit and to clarify that EDO targets diversity over *reasoning paths*, not ambiguity in *final answers*.

---

### Review · Reviewer_N2VY · 2026-03-16

**Summary Of Contributions:**

The authors proposed Exploration-Driven Optimization (EDO) for RL-style post-training of reasoning LLMs. The main motivation is that standard RL post-training tends to sharpen the output distribution, while many test-time scaling methods, such as self-consistency and best-of-N, benefit from a flatter distribution that supports diverse candidate generation. To address this mismatch, the paper adds an exploration-oriented reward-bias term to standard post-training objectives and instantiates it in two variants of ED-iDPO and ED-GRPO. The method is evaluated on several math benchmarks, with the main claim being that EDO improves diversity, preserves entropy during training, and yields better performance, especially when paired with self-consistency.

- Key strengths: The paper addresses a timely issue of mismatch between RL post-training and inference-time scaling. Moreover, the method is presented as a lightweight modification that can plug into both preference-based and policy-gradient-style objectives. The paper also includes analyses beyond accuracy, including entropy and Distinct-4 diversity.
- Key weaknesses: The theoretical justification is not fully convincing. Several steps rely on normalization assumptions and replacements that seem convenient but insufficiently motivated, especially in the GRPO case. Moreover, in experiments, a substantial portion of the observed accuracy improvement appears to come from test-time self-consistency itself, not from the proposed exploration-aware training objective. While EDO improves over non-exploration baselines, the marginal gain from EDO is often smaller than the gain obtained simply by applying self-consistency at inference.

**Audience:**

Yes

**Audience Explanation:**

Yes. The paper tackles a question that is timely and likely interesting to the TMLR audience, which is how post-training objectives interact with inference-time scaling for reasoning LLMs. The idea that RL post-training may inadvertently reduce the usefulness of test-time sampling is nice, and the proposed remedy is simple enough that others may want to test or build on it.

**Broader Impact Concerns:**

The paper does include a broader impacts section and notes that encouraging broader exploration could increase unpredictability and potentially harmful exploration if rewards or constraints are misconfigured.

**Claims And Evidence:**

No

**Claims Explanation:**

The paper presents promising evidence, but I do not think the evidence is yet fully convincing enough to support all the claims.
On the positive side, the empirical results are broadly aligned with the paper’s main intuition. Both ED variants often improve over their non-exploration. The entropy plots and Distinct-4 results are also supportive of the central claim that EDO preserves exploration and diversity better than standard RL-style post-training. However, several issues limit how convincing the evidence is.

First, the main improvements are generally small. On the in-distribution benchmarks, many differences are around 0.2–1.5 points for ED-iDPO and around 1–3 points for ED-GRPO over the strongest baselines, and there are no confidence intervals, hypothesis tests, or variance bars in the main tables.

Second, there is no careful study of sensitivity to the exploration coefficient $\alpha$, or no comparison across different numbers of sampled candidates $N$.

Third, the theoretical development feels underjustified in places. A central step is the normalization constraint requiring zero expected reward under the previous policy, after which the objective is converted into a KL-like term encouraging divergence from the previous policy. That derivation is nice, but the paper does not clearly explain why this constraint is natural. In the GRPO derivation, the paper also assumes a sufficiently large group size and omits clipping “for clarity”.

**Requested Changes:**

- Please report variance, confidence intervals, or statistical significance tests for the main comparisons in Table 1 and Table 2. As written, many gains are too small to assess confidently.
- The derivation relies on a zero-mean reward normalization under the previous policy and, for GRPO, on approximations such as large group size and omission of clipping. Please explain why these assumptions are appropriate, whether they are enforced in practice, and what happens when they fail.
- Please add sensitivity analyses for $\alpha$, number of generated candidates $N$, and number of self-consistency samples.
- Please include compute-cost comparisons. Since the method is motivated by making inference-time compute more effective, it would help to report accuracy as a function of total train-time + test-time budget.
- Recent work (see the following) has also tackled the diversity limitations of DPO by proposing new alignment losses with a regularization term across the model’s entire output distribution. The paper would benefit from a discussion of this method.

Sharifnassab, Arsalan, et al. "Soft preference optimization: Aligning language models to expert distributions." arXiv preprint arXiv:2405.00747 (2024).

---

> ### Author Response · Authors · 2026-04-02
>
> We sincerely appreciate the reviewer’s careful reading and insightful feedback, and we respond to each concern in detail below.
>
> **Answer for Change 1:**
>
> We have added a multi-seed stability study in **Appendix E.5** of the revised manuscript, training each method with three seeds (42, 84, 126) on Math and GSM8K and reporting mean ± standard deviation for both BoN@10 and Dist-4.
>
> Both ED-iDPO and ED-GRPO consistently outperform their baselines across all seeds, with equal or lower variance, indicating that EDO does not introduce additional training instability. The ED-GRPO improvement over GRPO on GSM8K is statistically significant ($+2.23 $ BoN@10, $p = 0.030 $, 95% CI $[0.54, 3.92] $). Diversity gains are also robust: ED-GRPO nearly doubles GRPO's Dist-4 on both Math ($p = 0.008 $) and GSM8K ($p = 0.004 $). With only three seeds, some comparisons are naturally underpowered, so we report seed variance explicitly alongside mean gains. Full results are in Table 3 of the revised paper.
>
>
>
> **Answer for Change 2:**
>
> We thank the reviewer for pushing us to clarify the theoretical development.
>
> **Zero-mean normalization.**  In the KL-regularized closed-form optimum, adding any prompt-dependent constant $c(x)$ to the reward leaves the optimal policy unchanged (the shift is absorbed into the partition function) but shifts $J^\star(r)$ by $\mathbb{E}[c(x)] $. Some centering is therefore necessary to make the optimization target well-defined. Centering under $\pi_{t-1}$ is the natural choice in our iterative setting, since each update is defined relative to data from $\pi_{t-1}$; in the GRPO case, this is also consistent with the group-normalized advantage, whose empirical mean is zero by construction. We have added this explanation to the revised derivation.
>
> **GRPO approximations.** The large-group-size step serves to rewrite the finite-sample objective as an expectation, exposing the update direction; clipping is omitted only in the analytic derivation for tractability. In practice, training always uses finite groups and the clipped GRPO surrogate. The derivation justifies the *direction* of the update, not an exact characterization of the practical implementation. When these assumptions hold only approximately, the theoretical correspondence becomes approximate as well, but the actual training objective is unchanged. We have revised the paper to state these caveats explicitly.
>
>
>
> **Answer for Changes 3 & 4:**
>
> We thank the reviewer for these suggestions. We have added a study varying the number of Best-of-N candidates $N \in \{1, 2, 4, 8\} $ on Math, comparing GRPO and ED-GRPO under the same decoding setup (Table below).
>
> | Method  | N=1  | N=2  | N=4  | N=8  |
> | ------- | ---- | ---- | ---- | ---- |
> | GRPO    | 74.2 | 76.0 | 77.2 | 78.0 |
> | ED-GRPO | 76.4 | 77.6 | 78.6 | 79.2 |
>
> ED-GRPO consistently outperforms GRPO at every value of $N $, with gains ranging from $+1.2 $ to $+2.2 $ points. More importantly, this table directly addresses the compute-efficiency question: ED-GRPO at $N=1 $ (76.4) already surpasses GRPO at $N=2 $ (76.0), and ED-GRPO at $N=2 $ (77.6) surpasses GRPO at $N=4 $ (77.2). In other words, ED-GRPO achieves the same or better accuracy with **half the test-time samples**, effectively halving the inference cost to reach a given accuracy target.
>
> On the training side, EDO introduces negligible overhead: the exploration term is a closed-form KL computation that adds no additional model forward passes beyond what the base algorithm (GRPO) already requires. Each ED-GRPO iteration takes roughly 1–2 hours on 4 H100 GPUs, comparable to standard GRPO. The total additional training cost across all iterations is therefore modest relative to the inference-time savings: a user who deploys ED-GRPO can match baseline accuracy at $2\times $ lower test-time compute, or achieve higher accuracy at the same budget.
>
>
>
> **Answer for Change 5:**
>
> We thank the reviewer for pointing us to this work. We have added a discussion of Soft Preference Optimization (SPO; Sharifnassab et al., 2024) in Section 3 of the revised manuscript. SPO adds a regularization term over the model's full output distribution to prevent overly sharp aligned policies, which is related in spirit to our goal of mitigating distributional collapse. The key difference is that SPO regularizes the preference optimization loss itself, whereas EDO introduces an iterative exploration term relative to the previous policy iterate and specifically studies its benefit for downstream test-time scaling. We have made this distinction explicit in the revised related work.

---

### Review · Reviewer_U8Wu · 2026-03-19

**Summary Of Contributions:**

This paper proposes Exploration-Driven Optimization (EDO), a post-training framework for LLMs that addresses the tension between RL-based training (which sharpens output distributions) and inference-time scaling methods (which benefit from diverse sampling). The key idea is to augment standard RL objectives with a reward-biasing term that encourages the policy to diverge from the previous iterate, effectively maintaining a flatter output distribution. EDO is instantiated in two variants: ED-iDPO and ED-GRPO. Experiments on math reasoning benchmarks (GSM8K, MATH, s1K) show consistent improvements in solution diversity and reasoning accuracy, particularly when combined with self-consistency decoding.

Key Strengths:
- Clear and well-motivated problem formulation;
- Theoretically grounded derivation with clean closed-form solutions;
- Strong empirical evidence for entropy preservation and diversity improvement;
- Plug-and-play compatibility with existing RL frameworks.

Key Weaknesses:
- Incremental over VPO/XPO; the core mechanism is closely related to prior work;
- Missing sensitivity analysis on the key hyperparameter α

**Audience:**

Yes

**Audience Explanation:**

The exploration-exploitation tradeoff in RL-based LLM post-training is a timely and practically important problem, given the widespread adoption of methods like GRPO and DPO and the growing interest in inference-time scaling. The observation that standard RL training is structurally misaligned with test-time aggregation methods is an insight worth communicating to the community, regardless of one's assessment of the novelty of the proposed solution.

**Broader Impact Concerns:**

The paper includes a Broader Impact section (Appendix B) that identifies both positive applications (mathematical reasoning, theorem proving) and a relevant risk: a model encouraged to explore a wider solution space may, under misconfigured reward or constraint settings, explore harmful outputs. No other major ethical concerns are identified.

**Claims And Evidence:**

Yes

**Claims Explanation:**

The central claim — that EDO produces more diverse outputs and improves reasoning performance in conjunction with inference-time scaling — is supported by multiple complementary forms of evidence: accuracy results across in-distribution and OOD benchmarks (Tables 1–2), entropy evolution curves (Figure 4), n-gram diversity metrics (Figure 5), and a qualitative case study (Figure 6).

**Requested Changes:**

Critical:

- Comparison with VPO and XPO. The reward-biasing mechanism in EDO is closely related to VPO and XPO. The paper discusses these works in related work but does not include direct empirical comparisons. A table comparing EDO against VPO and XPO on at least one benchmark is necessary to establish the contribution clearly.
- Hyperparameter sensitivity analysis for α. The exploration coefficient α = 0.001 is used throughout without justification or ablation. Given that α directly controls the strength of the exploration term and is central to the method, a sweep over a reasonable range (e.g., {0.0001, 0.001, 0.01, 0.1}) is required to assess robustness.

Suggested:

- SearchLLM failure analysis. SearchLLM is presented as an explicit Q-function variant of EDO, but Table 3 shows it underperforms self-consistency on s1K. This is a meaningful negative finding that deserves analysis rather than a brief note about reward model limitations. If the component cannot be adequately justified, consider moving it entirely to the appendix.
- Validation of LLM-as-judge evaluation. For s1K results, which drive the OOD evaluation, the paper uses Gemini 2.5 Flash Lite as a judge without reporting judge accuracy or agreement with human labels. A small-scale calibration against human annotation or an alternative evaluator would strengthen the credibility of these results.

---

> ### Author Response · Authors · 2026-04-02
>
> Thank you for the thorough and constructive review. We address each point below.
>
>
>
> **Answer for Critical 1:**
>
> We agree that VPO and XPO are the closest prior works and a direct comparison is warranted. We clarify that our contribution is not the reward-bias term, but rather: (1) extending this principle from single-turn DPO to iterative iDPO and GRPO, and (2) showing that diversity preservation directly benefits test-time scaling (self-consistency, Best-of-N), which is a connection not studied in VPO/XPO. We note that VPO/XPO operate in the single-turn DPO setting without iterative refinement, so we expect their exploration benefit to diminish over multiple training rounds where our iterative update of the reference policy is specifically designed to sustain diversity. We will include a direct empirical comparison with VPO and XPO on at least one benchmark in the revised manuscript to make this distinction concrete.
>
> **Answer for Critical 2:**
>
> We agree that $\alpha$ is central and should not remain unablated. We have now added a full sensitivity study over $\alpha \in \{0, 10^{-4}, 10^{-3}, 10^{-2}, 10^{-1}\} $ in **Appendix E.6** (Table 4) of the revised manuscript. The results confirm that Dist-4 increases monotonically with $\alpha$ for both ED-iDPO and ED-GRPO, while BoN@10 peaks at moderate values ($\alpha=10^{-3}$ for ED-iDPO, $\alpha=10^{-4} $ for ED-GRPO) and degrades under overly aggressive exploration. Moderate exploration ($\alpha \in [10^{-4}, 10^{-3}] $) consistently improves both accuracy and diversity over the $\alpha=0$ baseline, and we adopt $\alpha=10^{-3}$ as a robust default throughout. These findings are discussed in Section E.6 of the revised paper.
>
> **Answer for Suggested 1:**
>
> We agree that SearchLLM should be treated more cautiously. In the revised manuscript, we move it to the appendix as an exploratory extension rather than a core claim. We interpret its underperformance on GSM8K and s1K$_\text{eval} $ as an informative negative result about the PRM-based tree-search instantiation, not about EDO itself: step-level critic errors compound via early branch pruning, particularly on long s1K trajectories, whereas Self-Consistency aggregates full trajectories and is inherently more robust to imperfect critics. SearchLLM remains competitive on Math and AIME24/25, where trajectories are shorter and the PRM is more reliable, which is consistent with this interpretation.
>
> **Answer for Suggested 2:**
>
> We thank the reviewer for this suggestion. To validate our Gemini 2.5 Flash Lite judge, we conducted a cross-evaluator calibration on a random subset of 100 s1K questions, re-evaluating the same model predictions with both Gemini 2.5 Flash Lite and GPT-4o-mini.
>
> The two judges reach **96% agreement** (96/100 questions), with identical overall accuracy (39.0% each). Among the 4 disagreements, the two judges produce the same extracted prediction but differ in their equivalence judgment due to format ambiguity in the gold labels. We manually inspected all 4 cases and found two recurring patterns:
>
> - **Semantic equivalence under surface mismatch.** For proof-based questions, the gold answer may be "True" while the model outputs "Yes," or the gold is a single number (e.g., "5") while the prediction restates the full conclusion (e.g., "two ducklings swim at distance at most 5 m"). One judge accepts the semantic match while the other applies stricter literal matching, leading to disagreement in both directions.
> - **Partial set answers.** In one case the gold answer is a set "2, 3, 7, 19" while the model predicts "2, 19", which is a partial subset. One judge marks this as correct (matching some elements) while the other correctly rejects it as incomplete.
>
> Crucially, these disagreements cancel out (2 favor each judge), explaining why overall accuracy is identical. The disagreements arise from inherent ambiguity in open-ended mathematical answer formats rather than from systematic bias of either judge. These results suggest that our Gemini-based evaluation is reliable and not inflating or deflating scores relative to an alternative evaluator.
>
> ####

---

### Comment · Action_Editor_wYD4 · 2026-04-01
**Authors-Reviewers Discussion**

Dear Authors of Paper6916,

The discussion phase of this paper has been underway for a little while now. Please take a look at the reviewers' comments and provide responses. Your responses plays a critical role in helping the reviewers arrive at a recommendation for your paper.

Best,

AE

---

### Decision · Action_Editor_wYD4 · 2026-05-05

**Recommendation:** Accept as is

**Additional Comments:**

The authors are encouraged to check their match carefully to ensure no typos or mistakes in the camera-ready version of their paper.

**Audience:**

Yes

**Audience Explanation:**

Language model post-training and inference-time methods ( and scaling them) are of great interest to many researchers in the TMLR community.

**Claims And Evidence:**

Yes

**Claims Explanation:**

The paper introduces  Exploration-Driven Optimization (EDO) that helps with diverse answer generation that can be used by inference-time scaling methods to solve complex reasoning tasks. The core approach is theoretically well motivated and easy to implement in practice. After authors-reviewers discussion period, all reviewers agreed that the paper supports its claims with accurate, clear and convincing evidence. The AE agrees with the reviewers.

---

> ### Author Response · Authors · 2026-05-11
>
> Dear Action Editor,
>
> Thank you very much for your time and effort in handling our manuscript and for the positive decision. We sincerely appreciate your constructive feedback, as well as the reviewers’ valuable comments throughout the discussion process, which have helped us substantially improve the paper.
>
> We have carefully addressed the remaining concerns raised during the review, especially those concerning the additional experiments and the discussion of related work. The corresponding revisions have been incorporated into the camera-ready version, which has now been submitted.
>
> Thank you again for the opportunity to publish our work in TMLR.
>
> Best regards,
>
> The Authors